# Updating the Clinical Application of Blood Biomarkers and Their Algorithms in the Diagnosis and Surveillance of Hepatocellular Carcinoma: A Critical Review

**DOI:** 10.3390/ijms24054286

**Published:** 2023-02-21

**Authors:** Endrit Shahini, Giuseppe Pasculli, Antonio Giovanni Solimando, Claudio Tiribelli, Raffaele Cozzolongo, Gianluigi Giannelli

**Affiliations:** 1Gastroenterology Unit, National Institute of Gastroenterology-IRCCS “Saverio de Bellis”, Castellana Grotte, 70013 Bari, Italy; 2National Institute of Gastroenterology-IRCCS “Saverio de Bellis”, Castellana Grotte, 70013 Bari, Italy; 3Guido Baccelli Unit of Internal Medicine, Department of Precision and Regenerative Medicine and Ionian Area-(DiMePRe-J), University of Bari “A. Moro”, 70121 Bari, Italy; 4Scientific Director, Italian Liver Foundation, 34149 Trieste, Italy; 5Scientific Director, National Institute of Gastroenterology-IRCCS “Saverio de Bellis”, Castellana Grotte, 70013 Bari, Italy

**Keywords:** alpha-fetoprotein (α-FP), des-γ-carboxy prothrombin (DCP), hepatocellular carcinoma, screening, surveillance, algorithm

## Abstract

The most common primary liver cancer is hepatocellular carcinoma (HCC), and its mortality rate is increasing globally. The overall 5-year survival of patients with liver cancer is currently 10–20%. Moreover, because early diagnosis can significantly improve prognosis, which is highly correlated with tumor stage, early detection of HCC is critical. International guidelines advise using α-FP biomarker with/without ultrasonography for HCC surveillance in patients with advanced liver disease. However, traditional biomarkers are sub-optimal for risk stratification of HCC development in high-risk populations, early diagnosis, prognostication, and treatment response prediction. Since about 20% of HCCs do not produce α-FP due to its biological diversity, combining α-FP with novel biomarkers can enhance HCC detection sensitivity. There is a chance to offer promising cancer management methods in high-risk populations by utilizing HCC screening strategies derived from new tumor biomarkers and prognostic scores created by combining biomarkers with distinct clinical parameters. Despite numerous efforts to identify molecules as potential biomarkers, there is no single ideal marker in HCC. When combined with other clinical parameters, the detection of some biomarkers has higher sensitivity and specificity in comparison with a single biomarker. Therefore, newer biomarkers and models, such as the Lens culinaris agglutinin-reactive fraction of Alpha-fetoprotein (α-FP), α-FP-L3, Des-γ-carboxy-prothrombin (DCP or PIVKA-II), and the GALAD score, are being used more frequently in the diagnosis and prognosis of HCC. Notably, the GALAD algorithm was effective in HCC prevention, particularly for cirrhotic patients, regardless of the cause of their liver disease. Although the role of these biomarkers in surveillance is still being researched, they may provide a more practical alternative to traditional imaging-based surveillance. Finally, looking for new diagnostic/surveillance tools may help improve patients’ survival. This review discusses the current roles of the most used biomarkers and prognostic scores that may aid in the clinical management of HCC patients.

## 1. Introduction

Hepatocellular carcinoma (HCC) is the main type of primary liver cancer [1]. HCC is the third most common cause of cancer death worldwide, as it is the second most common cause of cancer [2]. The risk factors vary by region, but patients with any type of cirrhosis, particularly those between the ages of 40 and 60 in Western countries [1], are at high risk of developing HCC, with an annual risk ranging from 1% to 8% [3,4]. 

HCC affects 70–95% of patients with chronic liver disease (CLD), especially those with hepatitis B (HBV) or hepatitis C (HCV) virus infection [1,2,3,4], with the remainder caused by alcoholic liver cirrhosis (ALD) and the progressive form of non-alcoholic steatohepatitis (NASH), and less commonly by organochlorine pesticides, aflatoxins, Wilson’s disease, hemochromatosis, and alpha-antitrypsin deficiency [5]. Meanwhile, new etiological factors, such as metabolic liver disease, are becoming more relevant and must be considered separately. 

The observed decrease in HCC incidence in many countries, including China, is associated with the implementation of HBV vaccination and therapy [6], HCV treatment programs [6,7,8] or reduced aflatoxin exposure and may be mitigated or even overshadowed in the future by the recently rising prevalence of metabolic syndrome and NASH [1,3,5]. Figure 1 shows a depiction of a future possible scenario of prognostic scores in the context of HCC.

Current guidelines for HCC screening include every six months abdominal ultrasound (US), with or without serum alpha-fetoprotein (α-FP), in patients with cirrhosis and subgroups with chronic hepatitis B virus (CHB) infection [1,9,10,11]. Limited randomized clinical trial (RCT) data among Asian patients with CHB and numerous cohort studies consistently demonstrate that screening is significantly associated with early HCC detection, increased curative treatment receipt, and improved survival [10]. Specifically, a 2018 meta-analysis found that when combined with US, α-FP can increase sensitivity for early-stage HCC detection (63% versus 45% with US alone), and a modelling study found that US and α-FP were the most cost-effective screening strategy across the majority of simulations [12].

However, insufficient early HCC detection/prevention and suboptimal risk stratification methods, a lack of therapeutic remedies for those people detected at late stages, irregular implementation of curative therapies in clinical practice, and competing risks of mortality from underlying liver disease can all lead to increased HCC fatality in high-risk populations. Indeed, only about 30% of patients with HCC in Europe today are diagnosed at an early stage when curative treatments are available [1]. Similarly, patients with advanced-stage disease in the United States (USA) have a 5-year survival rate of less than 5%, compared to more than 70% in those with early-stage HCC [13].

In cirrhotic patients, lectin-bound α-FP (α-FP-L3), des-gamma carboxy-prothrombin (DCP), and other emerging biomarkers are frequently used as HCC surveillance tests. Based on the disease stage and aetiology, they do, however, perform differently in terms of early HCC detection [9]. Nonetheless, except for α-FP and DCP, which are recommended by Japanese HCC guidelines, none of the biomarkers have been validated in Phase III clinical trials and are used in clinical practice [14]. This is due to the high heterogeneity of HCC biology, in which changes in various biochemical pathways contribute to tumorigenesis and, as a result, diverse biomarker expressions [15,16].

Despite some flaws and pitfalls, the role of specific biomarkers as alternative or complementary diagnostic tools for the current standard of care for early-stage diagnosis of HCC is being intensively researched in this context (Figure 2) [5]. 

The following section discusses the contemporary role of the most common biomarkers and prognostic scores that could improve the clinical management of patients with HCC.

## 2. Materials and Methods

We conducted a literature critical review on relevant international English articles published between 1984 and 2022 using the PubMed database. The following keywords were used and linked by “AND” to select scientific papers: “marker”, “biomarker”, “diagnosis”, “prognosis”, “predictive scores”, “hepatocellular carcinoma” and “HCC”. The following sections discuss HCC biomarkers and prognostic scores, as well as their use in screening tests, diagnosis, and surveillance. 

We present current information derived from clinical trials, prospective and cohort studies, and meta-analyses. We rejected all unrelated articles from a total of 64,626 articles. We chose articles that discussed the clinical applicability of biomarkers. Finally, we gathered 166 articles.

## 3. HCC Biomarkers

Figure 3 depicts the timeline and number of articles published on the most validated biomarkers in HCC up to 2023.

### 3.1. Protein Biomarkers

#### 3.1.1. α-FP

The α-FP is a glycoprotein linked to the growth and development of HCC, and it induces malignant transformation of liver cells as well as proliferation, migration, apoptosis, and immune escape [17]. 

The α-FP is the only widespread used biomarker for HCC identification and surveillance, although it is not assumed to have appropriate screening performance characteristics. α-FP elevations can be definitively caused by a variety of illnesses, leading to false positive results, especially in patients with active CHB and CHC infections [18]. 

According to European guidelines from 2001, α-FP has a low sensitivity of 39–64%, and specificity of 76–97% (at its traditional cut-off of 20 ng/mL), and a positive predictive value (PPV) of 9–32% for early-stage diagnosis of HCC [18], which Stefaniuk P et al. confirmed in a review study in 2010 [19]. When the sensitivity drops to 22% at higher cut-offs of 200 ng/mL, the specificity increases [1]. Notably, serial α-FP value changes have been shown to be superior to single α-FP values in the detection of early-stage HCC [20,21,22].

The serum levels of α-FP demonstrated good accuracy in HCC diagnosis, with the 400 ng/mL threshold outperforming the 200 ng/mL threshold in terms of sensitivity and specificity, whether α-FP was used alone or together with US [22]. Nevertheless, approximately two-thirds of patients with HCC 4 cm showed α-FP levels less than 200 ng/mL, and approximately 20% of HCC do not produce α-FP [19]. 

However, despite its poor sensitivity and specificity [23], the dosage of α-FP concentration is one of the historical biomarkers most used for detecting HCC in at-risk groups such as cirrhotic patients [18,24]. 

Moreover, among other potential applications, α-FP has been also used to predict postoperative prognosis after surgical liver resection [25], as well as the likelihood of neoplasm recurrence in patients following liver transplantation [26]. 

Nonetheless, according to recent data, the optimal α-FP screening threshold may now be lower (as low as 12–20 ng/mL) due to a reduction in false positive cases due to increased antiviral therapy use [1,27]. 

Ultimately, because α-FP is insufficient as a screening test on its own, it is likely to play a role in the early detection of HCC when combined with other tests.

#### 3.1.2. α-FP-L3

α-FP-L3, or lens culinaris agglutinin-reactive α-FP, is a fucosylated glycoform of α-FP that was proposed about three decades ago as an early detection biomarker for HCC [28]. 

Notably, fucosyltransferase modifies the α-FP carbohydrate chain during HCC development, resulting in α-FP-L3 with a sensitivity of 40–90% and a specificity of over 90% for detecting HCC based on cohort characteristics [19]. The ratio of fucosylated α-FP to total α-FP is expressed as the percentage of α-FP-L3 [19,29,30,31]. Additionally, despite the fact that α-FP-L3 levels can be elevated in severe hepatitis [32], it appears to be a useful marker for detecting HCC in its early stages and predicting recurrence [33]. 

Specifically, the pooled sensitivity, specificity, and positive (PLR) and negative likelihood ratios (NLR) of α-FP-L3% for the diagnosis of early HCC in a 2021 meta-analysis [34] of 2447 patients from six heterogeneous studies were 34% (95% confidence interval (CI) 30% to 39%, *p* < 0.0001), 92% (95% CI 91% to 93%, *p* < 0.0001), 4.46 (95% CI 2.94 to 6.77, *p* = 0.0033), and 0.71 (95% CI 0.61 to 0.82, *p* = 0.0004). The diagnostic odds ratio (OR) was 6.78 (95% CI 4.02 to 11.44, *p* = 0.0074). The summary receiver operating characteristic’s area under the curve (AUROC) was 75% (95% CI 57% to 94%).

According to recent data from a Phase III cohort (n = 397) in the USA (Singal AG et al., 2022), α-FP-L3 had a sensitivity of 46.2% and a false positive rate (FPR) of 10% in the six months preceding the diagnosis of HCC [35]. A cut-off of 8.3% had a fixed FPR of 10% and a sensitivity of 40% for early-stage HCC [36]. 

Furthermore, measurements of α-FP-L3 before treatment with the highly sensitive method were more useful for the diagnosis and prognosis of HCC than measurements with the conventional method in patients with α-FP less than 20 ng/mL [34].

Overall, these findings indicate that α-FP-L3 does not perform well enough as a standalone biomarker for HCC, but it may be useful in a biomarker-panel-based screening strategy.

#### 3.1.3. DCP

Des-γ-carboxy-prothrombin (DCP) is immature prothrombin that lacks carboxylation to various glutamate residues [19]. Prothrombin is shaped after the γ-carboxylation of vitamin K-dependent propeptides. DCP is produced due to an acquired post-translational defect in malignant cells’ vitamin K-dependent carboxylase sequence [37]. Hence, DCP production does not increase in CLD or cirrhosis, although it is a potential marker for the early diagnosis of HCC [19,38]. 

However, DCP measurement has no prognostic value in cases of vitamin K deficiency or inhibition of vitamin K function (i.e., in subjects receiving dicumarol therapy), because its synthesis is also induced by vitamin K deficiency, resulting in false positives. As a result, DCP is also known as PIVKA-II (protein induced in vitamin K absence). Despite these limitations, many studies have shown that DCP has higher sensitivity (48–62%) and specificity (81–98%) than α-FP in distinguishing HCC from other CLD [19,39,40]. 

Liebman et al. [40] and Koike Y et al. [41] demonstrated that DCP can be used as a very specific diagnostic and prognostic marker in HCC patients in two multicentre and prospective studies involving 76 and 227 patients, respectively. DCP has subsequently undergone Phase II and early Phase III validation. DCP alone showed an AUROC of 0.72 in a Phase II study of 131 early HCC patients [31]. However, a Phase III study revealed low sensitivity in detecting early HCC (26.3%) with a fixed FPR of 10% [42]. 

The overall sensitivity, specificity, PLR and NLR of DCP for the detection of HCC were 67% (95% CI, 58% to 74%), 92% (95% CI, 88% to 94%), 7.9 (95% CI, 5.6 to 11.2), and 0.36 (95% CI, 0.29 to 0.46) in a 2012 bivariate meta-analysis of 20 publications with significant heterogeneity [43]. The area under the bivariate summary ROC curve was 89% (95% CI, 85% to 92%) [43]. Another similar meta-analysis later confirmed that DCP had moderate diagnostic accuracy in HCC [44]. 

Notwithstanding, other data have suggested that DCP may not significantly improve the discriminatory power of α-FP and α-FP-L3 in the early detection of HCC [45]. 

Additionally, Sagar VM et al. discovered that DCP levels correlated with treatment response in most patients across a variety of therapeutic modalities in a cohort of 141 UK patients and found that DCP levels were informative in 60% of cases among α-FP non-secretors [46].

#### 3.1.4. GP73

The type II Golgi-localized transmembrane protein known as Golgi protein 73 (GP73) is primarily expressed by cells of the epithelial lineage, though hepatocytes in a healthy liver also express it in small amounts [47]. GP73 expression significantly rises in the liver of HBV and HCV virus-infected individuals with cirrhosis [48], or focal nodular hyperplasia [49]. Numerous tumour cell types exhibit high levels of the serum GP73, which can also be used to diagnose HCC [50,51]. Even though the underlying mechanisms that lead to elevated GP73 levels are unknown, their roles in its secretion and potential contribution to HCC diagnosis are extremely important [52,53]. 

GP73 plays a role in the development of HCC through multiple mechanisms, by promoting the epithelial–mesenchymal transition in HCC cells, by interacting with EGFR to control the latter’s cell-surface recycling [54,55], or partly by targeting TGF-β1/Smad2 signalling [56]. 

Additionally, GP73 works with MMP2 or MMP7 in HCC cells to encourage their secretion and movement, which aids in the metastasis of HCC cells [52,53]. It has been also shown that GP73 and α-FP work in concert by increasing α-FP secretion via direct binding to α-FP, and that GP73 can aid HCC cells that express α-FP and its receptor in cancer progression and metastasizing [50]. Furthermore, extracellular α-FP and GP73 worked together to intensify the HCC cells’ malignant phenotype [50], decreasing patient survival rates [50,51]. 

Interestingly, GP73 levels can also be used to assess the efficacy of an anti-cancer treatment [57], and to choose patients determining the likelihood of potential complications following hepatectomy [58].

According to a 2009 Chinese study by Li X et al. [59], GP73 tests have higher sensitivity and specificity for early detection of HCC than α-FP (sensitivity, 62% vs. 25%, and specificity, 88% vs. 97%; *p* < 0.0001, respectively), and GP73 serum levels increased with the malignant potential of CLD [58].

Medium GP73 levels were higher in an Asian population of 124 patients with various forms of CLD (*p* < 0.001) than in healthy individuals and patients with other diseases for the diagnosis of HBV-related HCC [60]. In patients with HBV-related HCC, GP73 had higher sensitivity, specificity, and AUROC than α-FP (87.1%, 83.9%, and 92% vs. 48.4%, 96.8%, and 77%, respectively) [60]. 

Instead, a 2012 meta-analysis of eight studies discovered that GP73 was just as reliable as α-FP in diagnosing HCC regardless of the aetiology of CLD. In the included studies, the summary estimates for GP73 and α-FP in diagnosing HCC were as follows: sensitivity, 76% (95% CI 51% to 91%) vs. 70% (47% to 86%); specificity, 86% (95% CI 65% to 95%) vs. 89% (95 % CI 69% to 96%); DOR, 18.59 (95% CI 5.33 to 64.91) vs. 18.00 (95% CI 9.41 to 34.46); and AUROC 88% (95% CI 77% to 99%) vs. 86% (95% CI 84% to 87%) [61]. 

More recently, GP73 levels, among other biomarkers, were significantly higher in HCC compared with the other groups (CHC with/without cirrhosis and healthy subjects) in a 2020 Egyptian study of 238 patients (*p* < 0.001). For GP73, the ROC curve analysis revealed 91% sensitivity, 85% specificity, 74.7% PPV, and 95% NPV (AUROC 96%) [62].

Therefore, GP73 serum levels may be comparable to α-FP as promising HCC predictor biomarkers.

#### 3.1.5. GPC-3

Glypican-3 (GPC-3) is a glycoprotein that belongs to the proteoglycan family that contains heparan sulfate and is expressed in 72–81% of HCC cases [63]. 

GPC-3 serum levels are associated with a poor prognosis, as well as advanced tumour stage detection, vascular invasion, and metastases [64]. Moreover, a rapid increase in GPC-3 expression is also linked to the progression of precancerous lesions to HCC [65] Moreover, GPC-3 detection allows HCC to be distinguished from healthy liver tissue, benign lesions, and liver cirrhosis [66].

Concurrent detection of GPC-3 and α-FP improves test sensitivity and specificity, allowing for earlier diagnosis and reducing the risk of misdiagnosis [67]. El-Saadany et al. (2018) investigated the use of GPC-3 to aid in the diagnosis of HCC by comparing it to 20 healthy controls and two groups of 80 patients with α-FP less than or greater than 400 ng/mL. GPC-3 levels were significantly higher in HCC patients than in healthy controls [68]. 

As a result of the lack of clinical evidence on the reliability of GPC-3’s HCC diagnosis, we are presently unable to remark on its chances.

#### 3.1.6. OPN

Osteopontin (OPN) is a highly phosphorylated glycoprotein that either stays inside the cell or is secreted as an inflammatory cytokine. It is only expressed by liver macrophages, Kupffer cells, and stellate cells. OPN mediates a wide range of biological functions in the immune and vascular systems and has previously been evaluated as a tumour marker [69]. 

Increased OPN expression plays a pivotal role in hepatic inflammation, tumorigenesis, angiogenesis, extracellular matrix degradation, cancer cell migration, and metastatic potential in liver cancer and other digestive system neoplasms [70,71,72,73]. 

Multiple studies have found higher serum and plasma levels of OPN in people with HCC when compared to people with liver cirrhosis and/or CLD controls [74,75]. 

The researchers used Asian cohorts; a large multicentre study using West African and European cohorts was able to replicate these findings [76]. 

Across most studies, OPN had an AUROC ranging from 70% to 89% for predicting HCC. In contrast, the diagnostic efficacy of OPN in detecting early-stage HCC vs. non-HCC patients varied significantly depending on the study.

Shang S. et al. [77] and da Costa et al. [78] reported an AUROC of 73% and 0.70, respectively. In the latter case–control study from France, OPN at a cut-off level of 91 ng/mL, appeared to be effective in distinguishing CLD from HCC, whereas Ge T et al. reported an AUROC of 89% [75]. Interestingly, a prospective study of 115 Asian patients with CLD at risk of HCC revealed increased plasma OPN levels 24 months before diagnosis in 21 subjects who developed HCC [78].

OPN has also demonstrated promising results in the detection of α-FP-negative HCCs [74,79]. OPN and α-FP serum levels together better predicted HCC development than these markers separately [80]. According to a 2018 meta-analysis, the pooled sensitivity, specificity, and diagnostic OR for serum OPN were 81% (95% CI 67% to 90%), 87% (95% CI 77% to 93%), and 30.05 (95% CI 8.84 to 102.07), whereas 64% (95% CI 54% to 73%), 96% (95% CI 91% to 98%), and 41.52 (95% CI 13.69 to 125.93) for α-FP [81]. For the diagnosis of HCC, OPN showed a higher diagnostic accuracy than α-FP. 

Moreover, elevated OPN levels were associated with a worse prognosis and a shorter post-hepatectomy survival time because of HCC [82,83]. A meta-analysis involving 9150 patients in 2022 confirmed that OPN has the potential to be used as a promising predictive tumour biomarker in the early detection and prognosis of HCC [84]. 

As a result, combining α-FP and OPN can improve the sensitivity of early HCC diagnosis.

#### 3.1.7. Dickkopf-1

Dickkopf-1 (DKK1) is a member of the DKK Family and a secretory glycoprotein [85], that is tumour-specific and highly expressed in adult liver and other gastrointestinal neoplasms [86]. 

It is unclear exactly how DKK1 contributes to the development and progression of HCC. However, by altering the tumour microenvironment and causing inflammation, DKK1 seems to promote tumour invasion and migration via TGF-β1 [87]. Additionally, as recently shown, DKK1 stimulates HCC angiogenesis and tumorigenesis via VEGFR2-mediated mTOR/p70S6K signalling [88].

Originally, DKK1 expression was primarily examined in HBV-induced HCC. The optimal diagnostic cut-off value for DKK1 was 550.93 ng/L in a 2017 Chinese study, and the percentage of plasma DKK1 was significantly higher in the HCC group than in the HBV-related liver cirrhosis, CHB, and healthy controls (*p* < 0.05) [89]. Nevertheless, when compared to α-FP, DKK1 has been deemed as less effective in the diagnosis of HCC [90]. 

However, in a 2012 Chinese study [91] enrolling 831 participants, DKK1 levels were significantly higher in HCC patients than controls, and similar findings were found for early-stage HCC (AUROC 86%, sensibility 70.9%, and specificity 90.5% in the test cohort; 90%, 73.8%, and 87.2% in the validation cohort). Additionally, when compared with all controls, DKK1 maintained diagnostic accuracy in patients with HCC who did not have elevated levels of α-FP (AUROC 84, sensitivity 70.4%, and specificity 90% in the test cohort), AUROC 87%, sensitivity 66.7%, and specificity 87.2%. This included patients with early-stage HCC (AUROC 87%, sensitivity 73.1%, and specificity 90.0% in the test cohort; AUROC 89%, sensitivity 72.2%, and specificity 87.2% in the validation cohort), compared with controls [91]. 

Additionally, the authors of a 2016 Egyptian study used DKK1 as a biomarker for early HCC detection in HCV-infected patients [92]. A significant drop in DKK1 five days after curative resection in a small group of patients who had their HCV-induced HCC surgically removed may have indicated it as a surveillance marker for recurrence [92]. 

Moreover, DKK1 has been linked to HCC metastasis and prognosis [93]. Indeed, overexpression of DKK1 was linked to beta-catenin cytoplasmic/nuclear accumulation in clinical HCC samples (*p* = 0.011, correlation coefficient = 0.144) in a group of 314 Chinese HCC patients, as a critical indicator of a poor clinical outcome in HCC patients (*p* = 0.011, correlation coefficient = 0.144) [93].

Instead, in a 2019 Egyptian study, DKK1 had an AUROC of 83% with 87.3% sensitivity and 82.9% specificity in HCC (HCV-related) patients at a cut-off point of 8.92 ng/ml [78]. DKK1 was found to be associated with tumour size, liver dysfunction, and poor performance status in HCC patients [94].

Conclusively, DKK1 may support α-FP in the diagnosis and surveillance of HCC, aid in identifying patients with α-FP-negative HCC, and help differentiate between HCC and benign CLD. 

#### 3.1.8. AFU

Alpha-L-fucosidase (AFU), which has two isoforms, alpha-l fucosidase (AFU1) and AFU2, is an enzyme that can remove the terminal -l-fucose residues from glycoproteins. 

Interestingly, high alpha-l-fucose expression has been linked to a variety of cancers, including breast, thyroid, and colorectal cancers [95].

In the diagnosis of HCC, AFU has been shown to be a promising tumor marker particularly in patients with underlying viral hepatitis and cirrhosis [96].

A 2014 meta-analysis of 12 studies aimed at evaluating the diagnostic value of AFU for HCC found a pooled sensitivity of 72% and a pooled specificity of 78% for AFU. The AUROC value was 81%. As a result, as a serum marker, AFU was useful in the diagnosis of HCC [97].

Because of its poor diagnostic performance, it has been suggested that AFU be measured in conjunction with other biomarkers to improve HCC detection sensitivity.

#### 3.1.9. AXL

AXL is a potential serum marker for the diagnosis of HCC. 

The activation of hepatic stellate cells and modulation of hepatocyte differentiation by the receptor tyrosine kinase AXL and its ligand Gas6 is critical in the development of liver fibrosis and HCC [98].

When compared with healthy or cirrhotic controls, AXL outperforms α-FP in detecting very early HCC and has a high diagnostic accuracy in α-FP-negative patients [99,100].

Among α-FP-negative HCC patients with non-HCC patients, the cut-off was 1301 pg/mL (AUROC, 90%) with a sensitivity of 84.6%, a specificity of 76.3%. The optimal cut-off for AXL in differentiating all HCC and CLD patients was 1243 pg/mL (AUROC, 84%) with sensitivity 93.8%, specificity 61.9%. The combination of AXL and α-FP improved sensitivity for early HCC diagnosis [100]. 

#### 3.1.10. MDK

Midkine (MDK) is a heparin-binding growth factor that was discovered as a retinoic acid responsive gene and is involved in cell growth, survival, migration, angiogenesis, and carcinogenesis [101]. MDK expression has been found to be abnormal in a variety of human carcinomas, including HCC [102].

A 2019 meta-analysis discovered that HCC diagnostic accuracy of serum MDK was moderate/excellent [92,93]. The sensitivity and specificity of MDK for HCC diagnosis were 85% (95% CI 78% to 91%) and 83% (95% CI 76% to 88%), respectively [103,104]. A PLR of 5.05 (95% CI 3.33 to 7.40), a NLR of 0.18 (95% CI 0.11 to 0.28), a diagnostic OR of 31.74 (95% CI 13.98 to 72.09), and an AUROC of 91% (95% CI 84% to 99%) were also discovered. When the cut-off value was greater than 0.5 ng/mL, subgroup analyses revealed that MDK provided the best detection efficiency [103,104].

MDK levels were significantly higher in HCC compared to controls in a 2020 Egyptian study of 238 patients (*p* < 0.001). The ROC curve analysis revealed that MDK had 88.5% sensitivity, 80.6% specificity, 69% PPV, 93.5% NPV, and AUROC, 91%; MDK levels were comparable to α-FP levels in HCC patients [62]. 

Similarly, a 2020 systematic review and meta-analysis of 2483 patients discovered that MDK is more accurate than α-FP in diagnosing HCC, especially in early-stage HCC and α-FP-negative cases. The analyses for recognizing HCC using MDK and α-FP were as specified: 83.5 vs. 44.4% sensitivity, 81.7 vs. 84.8% specificity, and 87% vs. 52% AUROC. The analyses for identifying α-FP-negative HCC using MDK were as follows: sensitivity, 88.5%, specificity 83.9%, and AUROC, 91% [105].

According to a 2021 Egyptian study, patients with HCC (86 HCV induced) had significantly higher MDK levels than patients with liver cirrhosis and healthy controls (*p* < 0.001). At a cut-off value above 5.1 ng/mL, MDK levels discriminated between cirrhosis and HCC with a sensitivity of 100% and a specificity of 90% [106].

MDK appears to be a promising biomarker for early detection of HCC, particularly in α-FP-negative cases, but more research is needed to validate it.

#### 3.1.11. AKR1B10

Aldo-keto reductase family 1 member 10 (AKR1B10) was discovered to be over-expressed in many cancers from various organs after being isolated from HCC [107].

The over-expression of AKR1B10 in early stages of well and moderately differentiated tumours, as well as its down-regulation in advanced tumour stages, demonstrated that AKR1B10 may be a useful marker for HCC differentiation [108]. 

Over the last decade, AKR1B10, has emerged as a potential biomarker for the diagnosis and prognosis of HCC, with experimental studies demonstrating roles for this enzyme in biological pathways underlying the development and progression of HCC [109]. AKR1B10 also correlated with worse prognosis in HCC patients [110,111].

Serum AKR1B10 levels were found to be higher in patients with HBV/HCV-related HCC compared with patients with other liver disorders (*p* < 0.05). In early- and intermediate-stage HCC, AKR1B10 levels increased significantly more than in advanced- and terminal-stage HCC. At a cut-off value of 1.51 ng/mL, the sensitivity (81.0%) and specificity (60.9%) for HCC diagnosis with AKR1B10 were both high [112]. Indeed, AKR1B10 was found to be up-regulated in association with serum α-FP and to be an independent risk factor for HCC in CHC patients, implying a role in early-stage hepatocarcinogenesis [113]. AKR1B10 upregulation might play a role in the early stages of HBV-related hepatocarcinogenesis [114]. However, its high expression may predict a low risk of early tumour recurrence after liver resection in patients with HBV-related HCC [115]. 

Moreover, even after SVR, CHC patients with high levels of hepatic AKR1B10 had an increased risk of developing HCC [116]. 

A multicentre study [90] with 1244 participants found that serum AKR1B10 levels were significantly increased in HCC patients compared with those in non-HCC and were associated with α-FP, alanine/aspartate aminotransaminase, tumour size, vascular invasion, and TNM stage, with an AUROC of 87%, sensitivity of 72.7%, and specificity of 95.7% for the diagnosis of HCC, and these values were better than those of AFP (AUROC 82%, sensitivity 65.1%, and specificity 88.9%), and AKR1B10 exhibited a promising diagnostic value (AUROC 89%, sensitivity 71.2%, and specificity 92.6%), and a similar diagnostic performance was observed in AFP-negative early-stage HCC (AUROC 83.9%, sensitivity 63.4%, and specificity 90.7%).

The ratio of AKR1B10 messenger RNA levels in HCC versus non-tumorous tissues may predict prognosis after curative hepatectomy, with low expression in HCC tissue indicating a poor prognosis [117].

#### 3.1.12. ANXA2

Several studies have revealed Annexin A2 (ANXA2) expression characteristics and distribution have good diagnostic potential for HCC diagnosis [118,119]. Moreover, ANXA2 is found to promote cancer progression and therapeutic resistance [120]. 

ANXA2 showed an AUROC of 80% across the entire range of sensitivities and specificities, whereas AFP had an AUROC of 78%. Combining serum ANXA2 and AFP detection significantly improved diagnostic efficiency (96.52%) and negative predictive value (96.61%) for HCC [119]. 

In a 2015 study aimed at assessing the diagnostic role of annexin A2 (ANXA2) as serum marker for 50 HCC patients, Annexin A2 levels were significantly higher in HCC patients’ sera compared with CLD patients’ sera (*p* < 0.001). The AUROC for ANXA2 was 91% at a cut-off level of 29.3 ng/mL [121]. 

A highly significant difference in serum ANXA2 levels was found among 44 mostly HCV-positive Egyptian patients with HCC and CLD, as well as controls. The AUROC of ANXA2 was 86%; the cut-off value was set at 18 ng/mL, with a diagnostic sensitivity of 74% and a specificity of 88%, whereas the sensitivity and specificity of AFP at the 200 ng/dL cut-off value were 20% and 100%, respectively [122]. 

A 2019 study found elevated ANXA2P2 expression levels in HCC tissue compared to adjacent noncancerous tissue, as well as a poor prognosis for patients with high ANXA2P2 levels in HCC tissue [123].

Furthermore, ANXA2 expression, or co-expression with STAT3 proteins, has been linked to HCC recurrence and survival [124]. 

#### 3.1.13. SCCA and Neoangiogenesis Genes 

Squamous cell carcinoma antigen (SCCA), a serine protease inhibitor that is naturally present in skin, as well as immunocomplexes forms of SCCA and α-FP (SCCA-IC and AFPIC, respectively), have both been identified in HCC patients and have been suggested as potential useful markers for the detection of micro-metastases and for improving accuracy of HCC diagnosis of at-risk patients.

A 2005 Italian study by Giannelli G et al., looking into the expression of SCCA in tumoral and peritumoral tissues, in the serum of 48 cirrhotic patients and 52 HCC patients found that in comparison to cirrhotic samples, HCC samples had significantly higher SCCA serum levels. Additionally, HCC tumoral tissue had a significantly higher level of SCCA expression than peritumoral tissue [125]. 

The same author found an inverse relationship between SCCA levels and tumour size in another study conducted in 2007 involving 961 patients. The AUROC for SCCA and SCCA-IC in smaller HCCs was 70% and 69.4%, respectively. Together, the use of AFPIC, SCCA, and SCCA-IC allowed for the detection of 25.6% HCC in patients with α-FP levels under 20 IU/mL [126]. 

Beale G. et al. compared 50 patients with HCC-, ALD-, or NAFLD-related conditions with controls with NASH-related cirrhosis in a cross-sectional study on various biomarkers in 2008. With SCCA-1 showing no greater benefit for HCC surveillance than α-FP and DCP, the authors demonstrated that the best biomarkers for HCC surveillance may depend on the underlying cause of CLD [127]. 

In a 2009 study with 27 cirrhotic patients and 55 HCC patients (36.4% with a single nodule less than 3 cm and 63.6% with a single nodule more than 3 cm (or multifocal)), the latter two groups demonstrated significantly higher serum SCCA levels than cirrhotic patients (1.6 and 2.2 ng/mL vs. 0.41 ng/mL, respectively), as well as higher SCCA values in hepatic tissue in cirrhotic patients (1163.2 microm^2^ and 625.8 vs. 263.8 microm^2^). The SCCA expression was significantly higher in smaller HCC [128].

In 103 patients with CHC, a 2012 multicentre prospective Italian study found that those who responded to HCV antiviral therapy had higher levels of SCCA-IC than those who did not respond (238 AU vs. 149 AU, respectively). Hence, SCCA-IC was also proposed to be used as a prognostic indicator of a patient’s response to anti-HCV therapy [129]. 

Additionally, a 2016 multicentre prospective Italian study demonstrated that angiopoietin-2 (ANGPT2), delta-like ligand 4 (DLL4), neuropilin (NRP)/tolloid (TLL)-like 2 (NETO2), endothelial cell-specific molecule-1 (ESM1), and nuclear receptor subfamily 4, group A, member 1 (NR4A1), in particular, were the liver five-gene signature associated with neoangiogenesis that reliably detected rapidly growing HCCs and predicted HCC-related mortality in cirrhotic patients of various aetiologies [130].

#### 3.1.14. GS

Glutamine synthetase (GS) is a metabolic enzyme that catalyzes glutamine synthesis (a major energy source for tumour cells) and has been identified as a sensitive and specific indicator for the development of HCC [131]. GS has been proposed as a promising marker for distinguishing between malignant and benign hepatocellular lesions.

Early studies have shown that GS is a novel serum marker for early HCC, particularly in patients with low α-FP levels (less than 200 ng/mL) [132].

Di Tommaso L demonstrated that the GS sensitivity and specificity for detecting early HCC were 72% and 100%, respectively [133]. Liu P et al. [134] discovered that serum levels of GS were higher in HCC patients compared with liver cirrhosis patients and healthy controls, and the AUROCs of GS and α-FP for HCC diagnosis were 0.85 and 0.861, respectively, whereas the AUROC was 91% (sensitivity 81.9%, specificity 100%) for differentiating AFP-negative HCCs from healthy controls, and the sensitivity and specificity were 82.5% and 93% when combining GS with these findings suggest that GS could be a useful biomarker for HCC diagnosis, particularly in α-FP-negative cases.

### 3.2. Genetic Biomarkers

#### 3.2.1. MicroRNAs and Circulating Cell-Free DNA

Exosomes and small nanoparticles have recently been described as extracellular carriers of a plethora of molecules and cellular compounds, particularly microRNAs (miRNAs), produced by liver cells and non-parenchymal immune cells [135]. They are detected in plasma and represent a type of liquid biopsy used to detect HCC early.

Exosomes have been hypothesized to be a potentially useful liver biomarker due to their mechanism of transmitting effector molecules and signals between cells. Several recent reports on exosomal miRNAs have found that these particles are better biomarkers for the diagnosis and treatment of HCC than their serum-free counterparts [135].

MiRNAs may be involved in variant OPN expression by interfering with translation by binding OPN mRNA in 3′-untranslated regions [136]. Additionally, in HCC cell lines, MiRNA 181a was observed to reduce OPN expression, and this may endow HCC with metastatic properties [136]. 

According to a 2016 meta-analysis, serum miRNAs have a relatively high diagnostic accuracy for HCC diagnosis and can easily distinguish HCC from healthy subjects and those with CLD/cirrhosis [137].

Sun N et al. studied the extracellular vesicles chip performance from 36 patients with early-stage HCC and 26 controls with cirrhosis in a 2020 study, with a sensitivity of 94.4% and a specificity of 88.5% [138]. 

Specifically, exosomal miRNAs with clinical significance in the detection, prognosis, and, in some cases, as a therapeutic target of HCC include: miR-224, miR-21, miR-93, miR-1247-5p, miR-92b, miR-210-3p, miR-155, miR-665, miR-718, miR-122, miR-638, miR-125b, and miR-9-3p are all examples of microRNAs [139]. Von Felden J et al. [140] found 86% sensitivity and 91% specificity for detecting early HCC in 209 at-risk controls in a 2021 phase II case–control study (AUROC, 87%). The signature of 3-small RNA clusters was independent of α-FP (*p* < 0.0001), and a composite model yielded an AUROC of 93% [139]. 

Circulating tumour DNA, on the other hand, has a promising diagnostic potential in hepatocarcinogenesis. DNA methylation has been identified as an early-stage circulating marker that could be used to detect HCC in its early stages [141]. It is, however, insufficient on its own and should be combined with α-FP for HCC screening and detection [142].

Although several different methylation panels are currently under investigation, there has been limited data beyond Phase II to support clinical use of DNA methylation. In a Phase II validation case–control study, an algorithm called the multitarget HCC blood test (mt-HBT), which includes three methylated markers in combination with α-FP and sex, demonstrated 82% sensitivity for early-stage HCC, 87% specificity, and an AUROC of 91% [143]. In another Phase II study involving 122 patients with HCC and 125 patients with CLD, a further multi-analyte cell free DNA test HCC (HelioLiver) demonstrated early-stage HCC identification of 76% with a specificity of 91% [144]. 

#### 3.2.2. LncRNAs 

Long noncoding RNAs (lncRNAs) are important players in oncogenesis and tumour development, according to mounting evidence [145].

FOXD2AS1 was found to be an oncogene in HCC, upregulating ANXA2 expression in part by sponging’ miR206 [146]. 

Lung-cancer-associated transcript 1 (LUCAT1) has been identified in several human cancers, but its role in HCC is unknown. However, LUCAT1 in HCC promotes tumorigenesis by inhibiting ANXA2 phosphorylation [147]. 

Cancer susceptibility candidate 11 (CASC11) has been shown to play an important role in a variety of cancers, including HCC [145]. CASC11 promoted the progression of HCC by means of EIF4A3-mediated E2F1 upregulation, indicating CASC11 is a promising diagnostic biomarker for HCC [145]. 

Zinc finger E-box binding homeobox 1 antisense 1 (ZEB1-AS1) is an oncogenic regulator found in a variety of cancers. A study discovered that ZEB1-AS1 could decoy miR-299-3p and upregulate E2F1 expression, elucidating the functions and mechanisms of ZEB1-AS1 in HCC tumorigenesis and progression and providing novel biomarkers for HCC [148].

In total, 131 lncRNAs were found to be differentially expressed in α-FP-negative HCC, and two lncRNAs (LINC00261, LINC00482) demonstrated good diagnostic power under the ROC curve [149].

Furthermore, it was recently discovered that LINC01133 promotes HCC progression by sponging miR-199a-5p and interacting with ANXA2. In patients with HCC, LINC01133 CNV gain predicts a poor prognosis [150].

High levels of RNA-binding proteins (RBPs), including lncRNAs, were found to be detrimental to patient survival in 21 cancer types, particularly HCC. The researchers discovered that RBP gene expression is altered in HCC and that RBPs perform additional functions beyond their normal physiological functions, which can be stimulated or intensified by lncRNAs and affect tumour growth [151].

#### 3.2.3. DNA Mutation and Methylation-Related Biomarkers

HCC has an incredibly diverse and complex genetic landscape [152] that has been clearly defined over the past 20 years, and this includes homozygous deletions on chromosome 9 and high-level DNA amplifications on chromosomes 6p21 (VEGFA) and 11q13 (FGF19/CNND1) (CDKN2A). The majority of mutations (60%) affect the TERT promoter, which is linked to higher telomerase expression [153]. Along with low-frequency mutated genes (like AXIN1, ARID2, ARID1A, TSC1/TSC2, RPS6KA3, KEAP1, MLL2), TP53 and CTNNB1 are the next most common mutations, affecting 25%–30% of HCC patients. These mutations help define some of the key deregulated pathways in HCC [153]. 

Specific genetic and molecular programs involved in hepatocarcinogenesis have been clarified by recent technological advancements in next generation sequencing (NGS). The molecular landscape of HCCs with vascular invasion was also examined in a recent study, which discovered distinct transcriptional, epigenetic, and proteomic changes fuelled by the MYC oncogene. They demonstrated that MYC up-regulates the expression of fibronectin, which encourages HCC invasiveness [154].

The mechanisms by which cyclin dependent kinase inhibitor 2A (CDKN2A), a crucial regulator of immune cell functionality, promotes immune infiltration in HCC are still unknown. A 2018 meta-analysis revealed that CDKN2A promoter methylation was linked to an increased risk of HCC, played a significant part in the development of HCC, and may be useful as a triage marker for HCC [155]. Another study from 2021 found that CDKN2A expression may have influenced how tumour-associated macrophages are regulated and that it can be used as a prognostic biomarker to assess the prognosis and immune infiltration in HCC [156].

Additionally, 18% to 40% of HCC patients have been found to have CTNNB1 mutations. The metabolic regulation of the liver is greatly influenced by the oncogenic Wnt/catenin pathway, which is triggered by the mutated CTNNB1. The metabolic morphology of CTNNB1-mutated HCC is distinct, frequently cholestatic, and infrequently with steatosis [157]. The rate of CTNNB1 mutation detection in HCC patients was increased by combining analysis of ctDNA and tumour tissue [158].

A 2021 study identified some candidate diagnostic and prognostic biomarkers for AFP-negative HCC, providing the top ten hub genes, which included several protein-coding genes such as EZH2, CCNB1, E2F1, PBK, CHAF1A, ESR1, RRM2, CCNE1, MCM4, and ATAD2 [149].

On the other hand, because epigenetic alterations such as DNA hypermethylation or hypomethylation are thought to be early events in HCC onset, DNA methylation biomarkers can be used to detect HCC.

In a large cohort study of 1098 patients with HCC and 835 healthy controls, an effective blood-based diagnostic prediction model combining 10-methylation markers (cg10428836, cg26668608, cg25754195, cg05205842, cg11606215, cg24067911, cg18196829, cg2321194, cg17213048, and cg25459300) was established, demonstrating the potential for HCC diagnosis with high sensitivity and specificity [141]. Furthermore, the sensitivity and specificity of this model’s ability to detect HCC were higher than those of AFP.

The detection of six HCC-specific hypermethylated sites (cg23565942, cg21908638, cg11223367, cg03509671, cg05569109, and cg11481534) was found to be highly sensitive and specific (92% and 98%, respectively) in separating HCC from other tumour types [159]. This study involved patients with various cancer diseases. In combination, it is believed that circulating tumour DNA methylation markers are accurate for use in HCC screening, diagnosis, and prognosis.

### 3.3. Immunological Biomarkers 

The recognition of tumour cells by leukocytes has been described for many different types of tumours, changing the interpretation of how circulating immune markers function in the oncogenesis process [160]. 

Because cancerous lesions release cytokines and chemokines into the bloodstream, they can be detected in at-risk patients. This is extremely crucial during the development of HCC because the tumour typically develops in the context of chronic hepatitis, where the excessive immune stimulation caused by the presence of an inflammatory response in the liver may lead to further alterations in measurable immune markers during the progress of HCC [161,162].

#### 3.3.1. Chemokines

Chemokines are essential immune system response mediators because they aid in the activation and recruitment of leukocytes at acute inflammatory or harmed sites. Chemokines are also crucial in tumor progression. 

Chemokines and their receptors, such as the CXCL12-CXCR4 axis, the CX3CL1-CX3CR1 axis, and the CCL20-CCR6 axis, have received a lot of attention in research [163]. 

C-C motif ligand 4 (CCL4) and CCL5 bind to the same receptor, C-C receptor 5, which is expressed in effector and memory T cells, making this interaction important in the control of chronic viral infections [164]. 

In cirrhotic patients, high serum levels of inflammatory chemokines such as CCL4 and CCL5 indicate the presence of HCC. While CCL14 is a potential prognostic biomarker that influences cancer progression and is linked to tumor immune cell infiltration in HCC [163].

One study has examined serum levels of various chemokines in the context of HCC detection, and multivariate regression analysis revealed that serum CCL4 and CCL5 levels were higher in cirrhotic with HCC (N = 61) than in cirrhotic patients without HCC (N = 78), making them useful candidate diagnostic markers for HCC. CCL4 and CCL5 detection performance for HCC was similar, with an AUROC of 0.72 for CCL5 and relatively high sensitivity of 71% and specificity of 68% [165].

#### 3.3.2. IL-6

Kupffer cells, or resident hepatic macrophages, are considered tumour-associated macrophages of HCC and can produce a variety of cytokines, most notably interleukin (IL)-6, a pro-inflammatory marker, to promote HCC tumorigenesis [166]. In patients with benign liver disease or non-HCC tumours, serum IL-6 levels are not elevated [167]. Serum IL-6 levels were found to correlate positively with tumour size and a poor prognosis in HCC patients [168].

IL-6 has the potential to be a useful HCC tumour marker. In a 2008 study, the ROC curves used to distinguish HCC from cirrhotic patients only showed that IL-6 titres had higher discriminant power than AFP titres, with a cut-off value of 12 pg/mL (sensitivity 0.73, specificity 0.87, efficiency 0.8). The sensitivity, specificity, and efficiency rates for discriminant analysis on HCC and non-HCC subjects were 77%, 93%, and 88%, respectively [169]. In a 2013 meta-analysis of HCC screening, IL-6 was found to be comparable (*p* = 0.66) to AFP [170].

Higher serum IL-6 levels, in particular, were discovered to be an independent risk factor for HCC development in female CHC patients but not in male CHC patients [171]. 

Furthermore, higher serum IL-6 levels have been linked to an increased risk of HCC regardless of hepatitis virus infection, lifestyle factors, or radiation exposure. Obese people are more vulnerable to this link [172]. 

When IL-6 is associated with biomarkers such as AFP measurement, its diagnostic value increases. 

## 4. Combination of HCC Biomarkers

With the availability of newer biomarkers and their various combinations, it has recently been determined that a US-free approach is a viable option for the early diagnosis of HCC. Assays that combine multiple biomarkers will be clinically significant in HCC decision-making processes.

When combined with DCP and α-FP, α-FP-L3 may be useful in HCC diagnostics and screening tests [173]. For the past 20 years, the α-FP-L3 and DCP have been routinely used in Japan [174,175], and the combination of these biomarkers has increased the likelihood of early detection of small HCC [19,174,175]. In a 2008 Japanese systematic review, α-FP performed worse than DCP, and α-FP-L3 in terms of diagnostic OR (4.50 vs. 8.16 and 10.50) and AUROC in patients with HCCs 5 cm or smaller with CLD or cirrhosis as controls (65% vs. 69% and 70%). For α-FP, DCP, and α-FP-L3 the optimal cut-off values were 200 ng/mL, 40 mAU/mL, and 15%, respectively [176]. In a large 2011 study of 270 Japanese HCC patients with serum α-FP levels less than 20 ng/mL, the combined use of the α-FP-L3% (using the highly sensitive detection method) and DCP biomarkers detected 49% of all HCC patients with a size less than 2 cm [34]. In a 2015 retrospective study of 1255 CHB Korean patients, Seo SI et al. found that adding DCP alone to α-FP increased the sensitivity of detecting early HCC to around 75%, with a specificity of almost 90% [177]. In a 2016 German cohort, the combined three biomarkers demonstrated sensitivity and specificity of 85% (N = 304 HCC patients versus N = 403 controls) [15]. In a 2017 meta-analysis and validation study, Chen H et al. found that DCP with α-FP (sensitivity, 84%, specificity, 86%; AUROC, 89%) performed better than DCP (sensitivity, 76%, specificity, 92%; AUROC, 84%) or α-FP (sensitivity, 73%, specificity, 92%; AUROC, 84%) alone [178]. The pooled sensitivity and specificity for the α-FP, α-FP-L3, and DCP integrated biomarkers were 88% and 79%, respectively, in a 2020 meta-analysis of thirteen studies, whereas the AUROC was 91%, and the diagnostic OR was 28.33 [174]. In a Phase II study conducted by Piratvisuth T et al. (2022), the combination of α-FP and DCP demonstrated the best clinical performance for detecting early-stage HCC when compared to other biomarkers [179]. 

Further advances in genomics and proteomics platforms, as well as biomarker assay techniques, have resulted in the identification of a variety of novel biomarkers, including GP-73, GPC-3, OPN, and microRNAs, that have improved HCC diagnosis [71]. 

Notably, α-FP-L3 or GP73 can also be used to diagnose α-FP-negative HCC, and though their combination improves diagnostic accuracy and sensitivity [180]. Intriguingly, testing the levels of α-FP, α-FP-L3, and GP73 in venous blood samples from the sublingual vein of high-risk populations, was also effective as a screening test for HCC, with the added benefit of being a simple, and inexpensive test [69]. 

El-Saadany et al. (2018) discovered that GPC-3 plus α-FP was the most sensitive and specific test for the diagnosis of HCC, with both sensitivity and specificity of 98.5% [68]. 

The combination of DKK1 and α-FP demonstrated a better diagnostic yield than α-FP alone [90,181,182]. Indeed, in a 2016 Turkish study, which included 39 healthy controls, 54 patients with cirrhosis, and 40 consecutive HCC patients, the α-FP levels varied in each group and could be used to distinguish between them (*p* < 0.001). The combined use of DKK1 and α-FP increased the diagnostic yield, with a sensitivity, specificity, PPV, and NPV of 87.5%, 92.3%, 92.1%, and 87.8%, respectively. The DKK1 levels could help distinguish the HCC group from the cirrhosis and control groups (*p* < 0.001) [90]. In a 2015 Chinese study by Ge T et al., AUROC was higher (95% vs. 83%) and sensitivity was higher (88.76 vs. 71.91%) than that of α-FP alone, when α-FP, DKK1, and OPN were used as a panel in 390 participants. Additionally, this combination demonstrated a significant improvement in early-stage HCC patient diagnosis [75]. 

On ROC analysis, serum MDK levels had better sensitivity and specificity than OPN and α-FP levels in the diagnosis of HCC (98.4%, 97.1%, and 97%) vs. (96.2%, 95.3%, and 95%) in a 2017 Egyptian study enrolling 170 patients [183]. Furthermore, combined analysis of both MDK and α-FP yielded a similar diagnostic value in the diagnosis of HCC as combined analysis of both OPN and α-FP (98% vs. 97.5%) [183]. As a result, serum MDK and OPN levels were comparable to α-FP levels as potential HCC diagnostic biomarkers in HCV patients with liver cirrhosis. 

The ROC curve analysis for GP73, MDK, and DKK-1 in a 2020 Egyptian study of 238 individuals revealed (1) 88.5% sensitivity, 80.6% specificity, 69% PPV, 93.5% NPV, and (AUROC 91%) for MDK; (2) 93.6%, 86.9%, 77.7%, 96.5% for DKK-1; (3) 91%, 85%, 74.7%, 95% (AUROC 96%) [62]. The combination of GP73 and DKK-1 had the highest AUROC value (99%), with 97.4% sensitivity and 93.1% specificity, followed by the combination of GP73 and α-FP (98%). As a result, serum levels of GP73, MDK, and DKK-1 were comparable to α-FP as promising predictor biomarkers for HCC patients. The two-marker panel of GP73 and DKK-1 demonstrated the highest specificity and sensitivity [62]. Similarly, a 2021 Egyptian study found that using GP73, DKK-1, and α-FP together improved the sensitivity and specificity for the diagnosis of HCC compared to using each one separately [86]. Finally, according to a 2021 meta-analysis, the sum of sensitivity and specificity of α-FP with GP73 was 1.76 (*p* = 0.0001), the best among all panels including multiple biomarkers. Moreover, the sum of the triple biomarker panel of α-FP, α-FP-L3, and DCP was greater (1.64, *p* = 0.0001) than any double biomarker panel [184].

The combination of ANXA2 and α-FP improved diagnostic sensitivity (98% specificity, LR + 41, and 97.6% PPV). Follistatin combined with α-FP provided 92% specificity but only 50% sensitivity. As a result, serum ANXA2 is a promising biomarker for HCC, especially when combined with α-FP. Follistatin combined with α-FP may improve HCC diagnosis specificity [121].

The integration of miRNAs and α-FP has also a promising future [185]. Interestingly, in a 2021 meta-analysis, the sensitivity of circular RNAs for HCC diagnosis was 82% (95% CI 78% to 85%), and the specificity was 82% (95% CI 78% to 86%), compared to 65% (95% CI 61% to 68%) and 90% (95% CI 85% to 93%) for α-FP. The AUROC for circular RNAs was 89% (95% CI 86% to 91%) and 77% (95% CI 74 to 81) for α-FP. The combination of circular RNAs and α-FP showed a sensitivity of 88% (95% CI 84% to 92%), a specificity of 86% (95% CI 80% to 91%), and an AUROC of 94% (95% CI 91% to 96%) [186]. Consequently, circular RNAs are reliable and promising biomarkers for detecting HCC, and their combination with α-FP may improve diagnostic accuracy.

Several retrospective studies on biomarker panels for the diagnosis of HCC, combining blood biomarkers with patient characteristics, have been published. 

In 192 HCC patients, a panel consisting of miR-122+miR885-5p+miR-29b+α-FP reported a 1.0 AUROC in the diagnosis of HCC [187]. On a sample size of 1933 subjects, the panel with ten DNA methylation markers demonstrated an AUROC of 94% in the training set and 97% in the validation set, indicating high sensitivity and specificity in the diagnosis of HCC [141]. Another panel consisting of miR-3126-5p+miR-92a-3p+miR-107+α-FP reported a 99% AUROC in 155 subjects, which was useful in the diagnosis of HCC [188]. 

A summary table with the main cited biomarkers and their combinations as well as their algorithms can be found in Table 1 and Table 2. A summary with the main cited biomarkers (with their timeline)requiring additional validation can be found in the Appendix A).

## 5. HCC Surveillance: The Prevailing Worldwide Guidelines 

The American Association for the Study of Liver Diseases (AASLD), Asian Pacific Association for the Study of the Liver (APASL), and European Association for the Study of the Liver (EASL) guidelines recommend HCC surveillance in the following cases, regardless of aetiology (for Child-Pugh A-B) [1,3,4]: liver cirrhosis, Child-Pugh C listed for liver transplant, HBV carriers with a positive family history of HCC, Asian males aged >40 years, Asian females aged >50 years, and African males aged >20 years.

A 2021 Cochrane meta-analysis by Colli A et al. [189] recommended combining α-FP (threshold around 20 ng/mL) with US in most clinical settings for diagnosing HCC in people with CLD. This resulted from the direct comparison in 11 studies (6674 participants) that revealed a higher sensitivity of US (81%, 95% CI 66% to 90%) versus α-FP (64%, 95% CI 56% to 71%) with comparable specificity: US 92% (95% CI 83% to 97%) versus α-FP 89% (95% CI 79% to 94%). A comparison of six studies (5044 participants) revealed that the combination of α-FP and US had higher sensitivity (96%, 95% CI 88% to 98%) and comparable results than US (76%, 95% CI 56% to 89%). For resectable HCC (two studies), US outperformed α-FP, and their combination outperformed them both, with sensitivity reaching up to 89% and specificity reaching up to 87% [189]. Despite the heterogeneity of the included studies, the combination of α-FP with US demonstrated the highest sensitivity, with fewer than 5% of HCC occurrences missed and approximately 15% false positive results.

Six-monthly monitoring effectively reduced HCC mortality by 37% [11]. Furthermore, surveillance allows for earlier detection of HCC and more frequent curative treatments, as revealed by a retrospective analysis of 887 cases of HCC diagnosed between 2005 and 2010 in the United States National Veterans Administration [194]. Even in patients with CLD and advanced fibrosis (fibrosis F3), the EASL/European Organization for Research and Treatment of Cancer (EORTC) guidelines recommend HCC surveillance with half-yearly US [1,3,4]. According to the AASLD guidelines, cirrhotic of any aetiology and cirrhosis related to CHB patients with a specific ethnic origin, age, and genetic background should have a six-month US and α-FP surveillance [195].

However, there is some uncertainty regarding the monitoring of patients with cirrhosis related to CHC (fibrosis F3) and HCV clearance [1,3,4]. In this regard, the AASLD HCV guidelines recommend HCC surveillance in this subgroup, whereas the AASLD HCC guidelines recommend HCC surveillance only in the presence of metabolic liver cirrhosis [1,3,4]. Furthermore, in cirrhotic patients with HBV and/or HCV, the APASL guidelines recommend six-monthly US and α-FP surveillance [196]. The Japan Society of Hepatology (JSH) guidelines recommend six-monthly surveillance with US and α-FP, α-FP-L3, and DCP in cirrhotic and CHB/CHC patients [197]. It should be noted that HCC may occur even in non-cirrhotic NAFLD patients, for whom surveillance is not recommended due to annual incidence rates of less than 1% [4].

Survival after HCC diagnosis varies between about 18 months in Germany, according to a retrospective analysis of 1066 patients with HCC divided into two 6-year periods (N = 385; 1998–2003 and N = 681; 2004–2009) [198], and about 48 months in Japan [199]. On the other hand, survival in the Western countries today is still comparable to that of Japan in the 1980s. Specifically, in Europe, approximately 30% of all HCC patients receive curative treatment, whereas in Japan, more than 60% receive this treatment [199].

### 5.1. HCC Predictive Scores in CHB 

Many of the HBV-related CLD-specific scores were developed in Asian patients with CHB to stratify the risk of developing HCC during surveillance. Following the need for an accurate, precise, and easy-to-use score in clinical practice, especially in geographic areas where the burden of HBV infection is particularly high, these scores were validated in Caucasian cohorts, reporting acceptable performances. A panel made up of age, gender, α-FP, and DCP was able to predict HCC in Chinese CHB patients in 2925 subjects with an AUROC of 94% in the training set and 93% in the validation set [95].

However, most of the validation scores performed worse than the “training” cohorts [3].

Validation studies with Caucasian CHB patients revealed the following AUROC value ranges for the respective scores, including specific variables: GAG-HCC ((age, gender, HBV-DNA, liver cirrhosis): 74–86%), CU-HCC ((age, albumin, bilirubin, HBV-DNA, radiological cirrhosis): 62–91%), REACH-B ((age, gender, ALT, HBeAg status, HBV-DNA levels): 54–77%), and RWS-HCC ((age, gender, liver cirrhosis, α-FP): 85%). 

These findings suggest that certain population characteristics have a significant impact on the risk of developing HCC in CHB patients.

### 5.2. HCC Predictive Scores in CHC 

In comparison to CHB, there are few studies on predictive models of HCC risk scores in CHC patients, and their main limitation is the lack of validation cohorts in most of them. 

In 2016, a single European study on a French cohort of 1323 CHC patients aimed to develop an individualized score for the prediction of HCC (age > 50 years, previous alcohol abuse, low platelet count, GGT > upper limit normal, and of sustained virologic response [SVR]) [3]. This score, however, lacked an external validation cohort. 

Finally, evidence that the risk of HCC decreases significantly over time in patients who have obtained sustained virological response (SVR) emphasizes the need for new algorithms to tailor HCC surveillance.

### 5.3. HCC Predictive Scores in CLD 

The HCC risk scores identified in studies evaluating Caucasian cohorts in both the study population and the external validation group, regardless of the aetiology of the underlying CLD, are as follows [3].

The THRI (Toronto hepatocellular carcinoma [HCC] risk index) was developed to predict the 10-year risk of HCC using four variables (age, gender, aetiology, platelet count); its performance was studied in three external validation cohorts from the Netherlands, China, and Turkey with similar accuracy in predicting HCC development. In identifying the high-risk group of HCC, AUROC values ranged from 75% to 80%.

The aMAP, a four-variable model (age–male–ALBI–platelets), was developed from a training cohort of 3688 Asian patients and validated in nine cohorts with different aetiologies and ethnicities. The optimal cut-off for predicting HCC was determined to be 50, with a sensitivity of 85.7–100% and a NPV of 99.3–100%. The aMAP score showed excellent discrimination and calibration in assessing the 5-year HCC risk among all the cohorts irrespective of aetiology and ethnicity [200].

### 5.4. Clinical Significance of HCC Biomarkers and Algorithms

Ideally, the use of serum biomarkers with sufficient sensitivity and specificity could allow for the early diagnosis of HCC, avoiding the need for US surveillance. Other than α-FP, several serum biomarkers have been included in some scoring systems for the prediction of HCC development [3].

However, except for α-FP and DCP, which are recommended by Japanese HCC guidelines, none of the biomarkers have been validated in Phase III clinical trials and are used in clinical practice [14]. This can be explained by the high heterogeneity of HCC biology, in which changes in various biochemical pathways play a role in tumorigenesis [16]. Indeed, HCC patients exhibit a wide range of patterns of positivity for HCC biomarkers [15].

Therefore, researchers have investigated combining biomarkers with patient-specific risk factors and/or diagnostic tools.

### 5.5. Hepatocellular Carcinoma Early Detection Screening (HES) and Parametric Empirical Bayes (PEB) Algorithms 

El-Serag HB et al. [201] reported on the development and validation of a α-FP-based algorithm in a retrospective cohort of cirrhotic patients with active HCV in the national Department of Veterans Affairs Healthcare System in 2014. The Hepatocellular Carcinoma Early Detection Screening (HES) algorithm took the patient’s current α-FP level, rate of α-FP change, age, alanine aminotransferase level, and platelet count into account. When compared to α-FP alone, this α-FP-adjusted model improved predictive accuracy at various α-FP cut-offs within 6 months [201]. Specifically, based solely on 20 ng/mL α-FP, the probabilities of HCC were 3.5% and 11.4%, respectively. Patients with the same α-FP values (20 ng/mL and 120 ng/mL) who were 70 years old, with ALT levels of 40 IU/mL and platelet counts of 100,000 had 8.1% and 29.0% chances of developing HCC, respectively [201].

Tayob N et al., in 2018, used the same previous study population to develop a risk prediction model called the parametric empirical Bayes (PEB) algorithm, which incorporates routinely measured laboratory tests, age, and the rate of change in α-FP over the previous year, with the current α-FP [202]. The analysis cohort included 11,222 cirrhosis control patients and 902 HCC cases who had serial α-FP tests prior to their HCC diagnosis. The PEB algorithm had a higher early HCC detection sensitivity of 63.64% when compared with the current α-FP alone (53.88%) [202].

Tayob N et al. also conducted a 2019 validation study of the model presented [203] in which 4804 cases of HCC with cirrhosis of any aetiology, were evaluated on the long term at Veterans Affairs medical centres. Within six months before diagnosis, the HES algorithm had a modest superiority in identifying patients with HCC with 52.56% sensitivity compared to 48.13% sensitivity for the α-FP assay alone (*p* < 0.001). The authors estimated that the HES algorithm detected almost 199 HCC cases per 1000 imaging analyses versus 185 for the α-FP assay alone, resulting in the detection of 13 additional HCC cases (*p* < 0.001) [203].

### 5.6. GALAD Score for HCC 

The GALAD (Gender, Age, α-FP-L3, AFP and DCP) score combines serum-based markers (α-FP, α-FP-L3, and DCP) with demographic factors (gender and age).
GALAD score: −10.08 + 0.09 ∗ Age + 1.67 ∗ Gender + 2.34 ∗ log(α-FP) + 0.04 ∗ α-FP-L3 + 1.33 ∗ log(DCP)

The GALAD score is a model developed using data from 833 patients (394 with HCC and 439 with CLD of mixed aetiologies) from the United Kingdom [204]. The GALAD score demonstrated exceptional overall performance (AUROC values of 95%, sensitivity of 92%, and specificity of 85%), which was maintained in the early detection of HCC (AUROC of 92%, sensitivity of 92%, and specificity of 79%) using the UK cohort [204]. 

The model was validated in independent cohorts from Japan, Germany, and Hong Kong (n = 6834; 2430 HCC (1038 early stage), and 4404 CLD) after comparison with the set of UK data, with an overall sensitivity ranging from 80% to 91%, specificity ranging from 81% to 90%, and AUROC values ranging from 85% to 95% across the populations [205]. The aetiologies in both the original and validation studies were mixed, ALD and CHC dominating. According to the cohort analysis, the GALAD score had better ROC curves than single markers regardless of ethnic origin or aetiology (HCV, HBV, and others), detecting early-stage tumours as efficiently as late-stage ones as emerged from the cohort analysis German (n = 275 HCC, n = 900 CLD; unifocal HCC < 3 cm = AUROC 87%; unifocal < 5 cm = AUROC 85%; unifocal < 4 cm = AUROC 87%; unifocal < 10 cm = AUROC 90%) [205].

A 2019 Phase II validation study in patients with NASH-related cirrhosis and early-stage HCC from a multicentre German cohort had a sensitivity of 68% and a specificity of 95% [206]. The GALAD score was also compared to US for the detection of HCC and found to be superior.

In a Chinese study, Liu M et al. (2020) [207], evaluated GALAD performance and developed new models in a country where HBV is the leading cause of HCC. They created the GALAD-C model with the same five variables as GALAD, as well as the GAAP model with gender, age, α-FP, and DCP, using logistic regression on 242 patients with HCC and 283 patients with CLD and comparing results to patients with other malignant liver tumours and healthy controls (50 patients, respectively). GALAD-C and GAAP models performed similarly (AUROC, 92% vs. 91%), and both outperformed GALAD, DCP, α-FP, and α-FP-L3% (AUROCs, 89%, 87%, 75%, and 71%) for discriminating HCC from CLD. 

The GALAD, GALAD-C, and GAAP models had excellent AUROCs for the HCV subgroup (94%, 96%, and 94%, respectively), but were relatively lower for the HBV subgroup (85%, 89%, and 88%). In a cohort of Chinese patients, the GAAP and GALAD-C models performed better than the GALAD model. These three models performed better in the HCV subgroup than those with HBV [207].

In a 2021, retrospective German single-centre study by Schotten C et al. [208], in a Caucasian HBV/HCV cohort (182 patients with HBV and 223 with HCV), in the Barcelona clinic liver cancer (BCLC) 0/A cohort (7%), GALAD had a higher AUROC in distinguishing HCC from non-HCC, outperforming α-FP α-FP-L3 and DCP (94% vs. 86%, 83%, and 83%, respectively). GALAD achieved even a higher AUROC of 96% and 98% in the HBV and HCV populations, respectively. Furthermore, GALAD had a significantly higher specificity (89%) in HCV patients than α-FP (64%) alone in detecting early-stage HCC. Hence, the GALAD score was deemed potentially useful for HCC surveillance in Caucasian HBV/HCV patients.

Interestingly, in a 2021 Japanese study by Toyoda H et al. [209], on HCC patients on dialysis, the α-FP, DCP, and GALAD scores all had high predictive values for HCC, with AUROC values greater than 85%. This effectiveness was maintained when focusing on small HCC (≤3 cm or ≤2 cm) or early-stage HCC, as well as after propensity score matching of HCC and non-HCC patient characteristics. DCP and GALAD scores had excellent predictive abilities for HCC. In conclusion, measuring serum HCC biomarkers can supplement imaging studies in the surveillance of HCC in dialysis patients, reducing the likelihood of advanced HCC at diagnosis.

In Singal AG et al., 2022’s prospective multicentre study [35], 42 patients with cirrhosis (Child-Pugh A/B) developed HCC (57.1% early stage) over a median of 2.0 years. In comparison to single-time point GALAD (79%), α-FP (77%), and HCC early detection screening (76%), longitudinal GALAD had the highest AUROC for HCC detection (85%). The highest sensitivities for HCC detection were observed for single time point GALAD (72%) and longitudinal GALAD (64%), respectively, in patients with biomarker assessment within 6 months prior to HCC diagnosis.

Tayob N et al. [36], on the other hand, conducted a 2022 prospective cohort Phase III biomarker study in the USA with 534 patients, 50 of whom developed HCC (68% early HCC) and 484 of whom had negative imaging. GALAD had the highest sensitivity (63.6%, 73.8%, and 71.4% for all HCC, respectively, and 53.8%, 63.3%, and 61.8% for early HCC within 6, 12, and 24 months), but a FPR (i.e., accuracy) of 21.5% to 22.9%. The AUROC, however, was comparable between GALAD, HES, α-FP-L3, or DCP. GALAD score outperformed in terms of increased false positive results, although a significant improvement in sensitivity for HCC detection.

According to Huang C et al. (2022) [210], the GALAD performs exceptionally well in the early diagnosis, prognosis prediction, and risk monitoring of HCC in a cross-sectional and longitudinal multicentre case–control study of 1561 Chinese patients. They discovered that GALAD identified early-stage HCC at an AUROC greater than 85% and outperformed α-FP, DCP, α-FP-L3, and BALAD-2. Meanwhile, the GALAD score could divide HCC into two distinct subgroups based on overall survival and recurrence risk. The GALAD score could detect HCC 24 weeks (AUROC, 85%) or 48 weeks (AUROC, 83%) before clinical diagnosis.

The GALAD score, on the other hand, performed worse in small recent Phase III cohorts, with one demonstrating a sensitivity of 53.8% and another demonstrating a sensitivity of 30.8% at a FPR of 10% [35,36]. 

These findings imply that GALAD results vary depending on clinical and not clinical parameters, and new findings from other studies are therefore awaited.

### 5.7. GALAD Score in Italy

The α-FP, α-FP-L3, and DCP levels were measured in 98 Italian patients (44 CLD patients without HCC and 54 HCC patients with predominantly HCV/HBV aetiology) using a mTASWako self-analyzer i30 (highly sensitive microchip capillary electrophoresis test and liquid phase binding assay) [211]. Serum levels of α-FP, α-FP-L3, and DCP were significantly higher in HCC patients than in CLD patients (*p* < 0.0001). The respective AUROCs values were 88%, 87%, and 87%.

The AUROC of the three combined biomarkers was lower than that of the GALAD model (98% vs. 95%, *p* = 0.028) [211]. According to the findings of this Italian study, the combination of α-FP, α-FP-L3, and DCP was superior to a single biomarker in detecting HCC.

Furthermore, the GALAD algorithm performs significantly better than the combination of these three biomarkers alone.

### 5.8. GALAD Score for NASH-Related HCC 

NAFLD is quickly becoming the leading cause of CLD, HCC, and liver transplantation. 

A multicentre case–control study conducted in eight German centres evaluated the GALAD score’s ability to diagnose HCC in patients with NASH on 125 patients with HCC (20% within the Milan Criteria, BCLC A) and 231 patients without HCC (NASH controls). The researchers also looked at data from a pilot cohort study of 389 NASH patients who were being monitored for HCC for a median of 167 months in Japan [206]. 

The GALAD score had significantly higher AUROC than the values for serum α-FP alone, α-FP-L3, or DCP (96% vs. 88%, 86%, and 87%, respectively). The AUROC values for the GALAD score were consistent in patients with and without cirrhosis (93%, and 98%, respectively). The GALAD score achieved an AUROC of 91% for the detection of HCC using the Milan Criteria, with a sensitivity of 68% and a specificity of 95% at a cut-off of −0.63 [206].

The mean GALAD score was higher in patients with NASH who developed HCC than in those who did not develop HCC as early as 1.5 years before the diagnosis of HCC in the Japanese cohort pilot study [206]. As a result, the authors conclude that the GALAD score can detect HCC with high accuracy in patients with NASH, both with and without cirrhosis as similarly to that of viral hepatitis. The GALAD score can detect patients with early-stage HCC and may aid in the surveillance of patients with NASH, who are frequently obese, limiting the sensitivity of US detection of HCC [206].

In any case, an optimal threshold value must be defined to develop a shared approach and standardized surveillance algorithm in patients with NASH.

### 5.9. GALADUS Score for HCC 

A recent study (2019) by Yang JD et al. found that the GALAD score outperformed the US in the detection of HCC [212]. However, the addition of a new GALADUS score (because of the combination of GALAD and US scores) improved the GALAD score’s performance even more. 

We looked at a single-centre cohort of 111 patients with HCC and 180 controls with cirrhosis or CHB from the Early Detection Research Network’s (EDRN) Phase II study, as well as a multicentre cohort of 233 patients with early HCC and 412 patients with cirrhosis from the EDRN Phase III study. The AUROC of GALAD for the detection of HCC was greater than that of US (85% vs. 82%, *p* < 0.01). The GALAD score had a sensitivity of 91% and a specificity of 85% at a cut-off of −0.76. The AUROC of the GALAD score for detecting early-stage HCC was 92% (cut-off −1.18, sensitivity 92%, specificity 79%).

In the EDRN cohort, the AUROC of the GALAD score for HCC detection was 88%. The GALADUS score improved the GALAD score’s performance in the monocentric cohort, reaching an AUROC of 98% (cut-off −0.18, sensitivity 95%, specificity 91%) [212].

The new model’s equation, known as the GALADUS score, is as follows:GALADUS = −12.79 + 0.09 ∗ age + 1.74 ∗ (1 for males, 0 for females) + 2.44 ∗ log10(α-FP) + 0.04 ∗ α-FP-L3 + 1.39 ∗ log10(DCP) + 3.56 ∗ (1 for positive US, 0 for negative US)

### 5.10. Novel Scores for Early HCC

By analyzing serum samples from 267 patients with liver cirrhosis, in 2021 Lambrecht J et al. [213] tried to develop a novel blood-based scoring tool for the identification of early-stage HCC. They created the APAC score, which consisted of the parameters age, expression levels of soluble platelet-derived growth factor receptor beta (sPDGFR), α-FP, and creatinine and identified patients with HCC among cirrhotic with an AUROC which was significantly better than the GALAD score (95% vs. 90%, *p* = 0.0031). Furthermore, the APAC score’s diagnostic accuracy was independent of disease aetiology. The APAC score achieved an AUROC of 92% (sensitivity 85%, specificity 89%) and 95% (sensitivity 91%, specificity 85%) for detecting patients with (very) early (BCLC 0/A) HCC stage or within Milan criteria, respectively. As a result, the APAC score was a highly accurate serological tool for the early detection of HCC.

Notably, the multitarget HCC blood test (mt-HBT), incorporating methylation biomarkers (i.e., HOXA1, TSPYL5, and B3GALT6), α-FP, and sex, demonstrated 72% sensitivity for early-stage HCC at 88% specificity in a 2022 multicentre US study by Chalasani NP et al. [143], with 136 HCC cases (60% early-stage) and 404 controls. An independent cohort of 156 HCC cases with 50% early-stage and 245 controls was used to validate the test performance, which showed 88% overall sensitivity, 82% early-stage sensitivity, and 87% specificity. Early-stage sensitivity in clinical validation was significantly higher than α-FP at 20 ng/mL or greater (40%; *p* = 0.0001) and GALAD at −0.63 or greater (71%; *p* = 0.03), despite α-FP and GALAD having higher specificities (100% and 93%, respectively) at these cut-off values. 

Finally, in a recent prospective, multicentre, case–control Phase II study by Piratvisuth T et al., the researchers looked at biomarkers mentioned in international guidelines for HCC surveillance and diagnosis to find combinations with high sensitivity and specificity for early-stage HCC in a group of patients with HBV, 32.9% HCV, 60.5% cirrhosis, and 40.6% with early-stage HCC. The combination of α-FP and DCP, and either published biomarker (i.e., IGFBP3, COMP, or MMP3), as well as age and gender, demonstrated the best clinical performance for detecting early- and late-stage HCC. These new panels’ performance was comparable to the GALAD score [179].

A summary table with the main cited algorithms can be found in Table 3.

## 6. Discussion

Because the majority of HCC patients are still diagnosed at advanced stages when locoregional treatments are no longer indicated [214], in this review we discuss recent advances in diverse HCC biomarker evaluation for HCC early detection and prognostic scores in at high-risk populations, as well as the difficulties and advantages of shifting beyond US-based HCC screening.

Cirrhosis incidence increased slightly in Europe, Asia-Pacific, East Asia, and Southeast Asia between 2000 and 2015 [215]. According to data from a Canadian population-based cohort, the incidence of age-specific cirrhosis increased by 22% between 1997 and 2016 [214]. In 2017, 1.5 billion people worldwide had CLD, with the most common causes being NAFLD (60%), HBV (29%), HCV (9%), and ALD (2%) [214]. Cirrhosis affects half of all North American “baby boomers” (born 1945–1965), with Black and Hispanic people, and those with lower education levels having a higher prevalence, and incidental diagnosis increasing in younger Americans [215]. 

As a result of large-scale HBV vaccination and HCV treatment programs, the rising prevalence of metabolic syndrome, a concurrent increase in the use of injectable drugs, and an increase in alcohol abuse, the epidemiology of CLD and liver cirrhosis, and thus of HCC, is changing [215]. Aflatoxin consumption with food (especially in East Asia and Sub-Saharan Africa), hemochromatosis, α-1-antitrypsin deficiency, and tobacco use remain significant risk factors for HCC in the minority of cases [215].

Accordingly, HCC is more prevalent in Asia and Sub-Saharan Africa than in Europe and North America, which has been linked to the spread of viral hepatitis [2,3,4]. There were approximately 854,000 new cases of liver cancer in 2015 (a 75% increase since 1990) and 810,000 cancer-related deaths worldwide [215]. From 1999 to 2017, there were approximately 150,000 HCC-related deaths in the USA. Males, Asian/Islanders, Pacific people, and those aged 75 to 84 years had the highest mortality rates in 2017 [215]. Although mortality rates have decreased in East Asia, North Africa/Middle East, and high-income Asia-Pacific, deaths have increased in many other parts of the world, including South Asia, Central Asia, and Eastern Europe. 

Racial disparities in HCC mortality persist, which research suggests is due, in part, to treatment failures, lower rates of early diagnosis, and lower chances of curative treatment, including liver transplantation, among racial minorities and non-Hispanic whites [215]. China has currently the most HCC cases due to both an elevated rate (18.3 per 100,000 people) and the world’s largest population (1.4 billion people) [216]. However, in recent years, there has been an increase in the incidence of HCC in patients with non-viral aetiology, primarily of a metabolic nature [1,2,3]. 

In Italy the majority of HCCs are caused by CHC and ALD, followed by CHB and others, despite a recent significant increase in NASH/NAFLD-related HCC (https://www.registri-tumori.it/cms/pubblicazioni/i-numeri-del-cancro-italia-2020 (accessed on 20 November 2022)) [1]. Additionally, liver tumours are more common in the south of Italy in women, and in the centre of Italy in men. According to the Italian Cancer Registries Association, 33,800 Italian men were diagnosed with liver cancer in 2020. It is estimated that approximately 13,000 new cases of liver tumours will be diagnosed in the same period (Males/Females ratio, 2:1), with 7800 deaths and a net survival rate of 21% in males and 20% in females.

The current international guidelines for HCC surveillance recommend US with or without α-FP every six months as the standard of care, which has been shown to be noninferior to three monthly intervals [217]. The sensitivity of the US for early-stage HCC was reported in a meta-analysis of cohort studies. The detection rate was only 45%, but with the addition of α-FP, it rose to 63% [211]. Nevertheless, this strategy has limitations in many countries worldwide due to patient characteristics (e.g., body habitus, NAFLD) as well as US operator experience [12,218], or diminished adherence of at-risk patients to US screening [219]. 

The other disadvantage of this strategy is the two-point assessment required, which includes radiologic imaging and a blood test, highlighting the need for more accurate single-point screening tests. However, a recent cost-effectiveness analysis that considered both the benefits and costs of HCC surveillance in patients with compensated cirrhosis found that US and α-FP are more convenient for surveillance than US alone or no surveillance at all [220]. 

Moreover, due to the biological diversity of HCCs, some people have normal α-FP levels but high α-FP-L3 or DCP levels, and vice versa. US exams are commonly performed and interpreted in real time by hepatologists and are regularly combined with the broad application of a variety of biomarkers such as DCP, α-FP, and α-FP-L3 [179]. According to the AASLD, it is critical to determine whether serum biomarkers, such as α-FP-L3, DCP, and other novel serum tests, complement the US. 

Presently, there are no conclusive results regarding the possible different influence of ethnicity on the outcome of biomarkers/algorithms for HCC. 

Colli A et al. discovered that between 1982 and 2020, there were no differences in sensitivity and specificity of α-FP (cut-off of 20 ng/mL) for detecting HCC between countries (Europe, 60% and 87%; America, 56% and 89%; Asia, 60% and 83%; Africa, 68% and 81%, respectively; *p* = 0.447) in 98 studies conducted in Asia, 22 in Europe, 7 in Africa, 19 in North and South America, and one in all three continents [189]. 

Zhou JM et al. included four Asian case–control studies (China, Taiwan, and Japan) and two Western (Germany and the United States) in a meta-analysis on α-FP-L3% performance for early HCC diagnosis, with diagnostic cut-offs ranging from 3.84% to 10%. Among Asians, sensitivity ranged from 13.3% to 52.6%, with specificity ranging from 84.2% to 98.4%, whereas in Western countries, sensitivity was lower (27.9%), with similar levels of specificity (91.3–96.9%) [190].

In the Gao P et al. meta-analysis evaluating DCP diagnostic performance for HCC detection, despite the difficulty of comparing six Caucasian and fourteen Asian studies because biomarker thresholds varied across studies (4.5–20.24 ng/mL, and 40–150 mAU/mL), the sensitivity of studies on Asian subjects was lower than that of Caucasian studies (63% vs. 76%), whereas specificity showed the reverse (93% vs. 88%). Nonetheless, these findings may be due to higher DCP values in Caucasian subjects without liver disease [43]. 

In a meta-analysis on the performance of GP73 using different cut-offs ranging from 4.36 ug/L to 400 ug/L, the four Chinese studies outperformed the four American and Turkish studies in terms of sensitivity (75–98% vs. 7–95%) and specificity (52–97% vs. 35–95%) [61].

Another meta-analysis to evaluate the HCC diagnostic performance of OPN found that the Western studies had slightly better sensitivity and specificity (75–100% vs. 26–90%; 66–100% vs. 64–93%, respectively) from seven Eastern studies (three Korean, two Chinese, one Thai, one Australian) and five Western studies (four Egyptian, two American).

In a systematic review and network meta-analysis involving 9080 patients and evaluating DKK-1 sensitivity and specificity for HCC detection, the authors did not conduct ethnic subgroup analyses. However, most of the studies were Chinese, with only two studies conducted in Egypt [181].

Finally, a meta-analysis [105] comparing the accuracy of MDK for HCC detection from nine Eastern studies (4 Chinese, 5 Australian) and five Egyptian studies found that the latter population had slightly better sensitivity and specificity (China (86–87% and 84–90%), Australia (61–79% and 58–63%), vs. Egypt (82–100% and 83–97%), respectively). However, it should be noted that the etiology of CLD differed by country (HCV is extremely common in Egypt) and by the fact that the studies used different HCC diagnostic cut-offs ranging from 0.387 ng/mL to 1.683 ng/mL.

Even though extensive clinical evidence supports the GALAD model’s superiority for HCC detection in Asian HBV and HCV cohorts, its use in Caucasian populations is still limited. A contemporary prospective study enrolled 6834 patients (2430 with HCC and 4404 with CLD) from Germany, Japan, and Hong Kong [205]. Despite no ethnicity subgroup analysis being performed, the sensitivity of the GALAD model for HCC was higher in the UK and Germany than in Japan, whereas the specificity and accuracy were similar (sensitivity (88.4–91.6% vs. 81.4%); specificity (88.2–89.7% vs. 89.1%); AUC (0.94–0.97 vs. 0.93)). There was no evidence of a difference in model performance between etiologies in the European cohorts. Although there was some proof of an etiological difference in the Japanese cohort, it was not clinically influential.

Moreover, while a US-based strategy was successful in some settings, such as Japan, where screening methods incorporating the α-FP, α-FP-L3, and DCP biomarkers allowed for the detection of a significant percentage of tumours measuring less than 2 cm [221,222], it may be challenging to spread in others. Likewise, as a result, more than half of the Japanese with a diagnosis of HCC had undergone surgery or local ablation therapy, implying that increasing HCC screening uptake in other contexts may spare meaningful national public health resources. 

The pursuit of a single-stage test has resulted in increased research into newer biomarkers and their combinations, as well as newer radiologic imaging [218]. 

Because α-FP, α-FP-L3, GP73, and OPN are insufficient as standalone biomarkers for early detection of HCC and DCP, DKK1, and MDK seem promising among α-FP non-secretors, they may be helpful in a biomarker panel-based screening strategy. On the other hand, clinical evidence still requires further studies, particularly concerning the reliability of GPC-3, SCCA, SCCA-IC, serum miRNAs, and cell-free DNA. Despite their various combinations that may enhance the diagnostic accuracy of HCC in its early stages, currently, none of these biomarkers are recommended due to a lack of RCT and cost–benefit analyses.

Previous and recent research has shown that the GALAD score has a higher sensitivity to both US and α-FP alone, as well as the combination of its component biomarkers [35,212]. Indeed, GALAD score increased sensitivity is associated with high rates of false positivity and 6–12 months pre-diagnosis, resulting in morbidity in terms of anxiety and cost due to additional tests to confirm the diagnosis [35,36]. Moreover, the same superiority was not demonstrated in the Phase III biomarker study in the USA by Tayob N et al. when the FPR was reduced to 10% [36]. 

Furthermore, while combining GALAD with US may alleviate these flaws, it defeats the purpose of single-point screening [33]. Moreover, while α-FP-L3 is not universally available and has a high cost of surveillance, the combination of α-FP and DCP has shown higher diagnostic accuracy than individual biomarkers and may be a viable strategy in viral hepatitis endemic countries [223].

Nevertheless, GALAD may still be an effective tool in HCC screening and surveillance, especially in diagnosing a subset of patients with a small tumour and negative α-FP and DCP [210,222], but it may not be the best strategy in the other cases [218]. Indeed, Sachan A et al. recently commented in a letter that, in the absence of clear superiority to current surveillance strategies, as well as the limitations of using the GALAD score universally in all HCC patients, its routine use for HCC screening and surveillance may no longer be feasible [218]. 

The addition of multiple biomarkers raises the cost of surveillance in a resource-constrained setting, and no cost–benefit analysis for the GALAD score is currently available. Similarly, the morbidity associated with a high FPR of 25% with the GALAD score versus 15% with the US and α-FP combination is concerning [36,189].

Additionally, NAFLD is quickly gaining importance as the leading cause of HCC, CLD, and liver transplantation even in the absence of liver cirrhosis [224]. According to the American Gastroenterology Association expert review published in 2020, only NAFLD patients with cirrhosis or advanced fibrosis as determined by non-invasive tests should be considered for HCC surveillance [225]. 

US surveillance is recommended, but it is limited in obese patients and those with NAFLD who are at high risk of having an inadequate imaging resolution. Of interest, GALAD score may play a keystone role in identifying NASH patients at high risk of developing HCC, where the role of US and DCP is still unknown [206].

Therefore, we emphasize the need for evaluation of the utilization of current and novel biomarkers in validation studies, paving the way for a shift away from a US-based paradigm for HCC screening, with significant potential benefits to patients in terms of reducing the burden of HCC in at-risk populations. 

## 7. Conclusions

In conclusion, more sensitive tests that work around these current barriers to HCC surveillance adherence are required due to several limitations of the US-based screening strategy. In particular, there is a chance to produce and validate better data supporting the use of HCC screening and prognostic scores as an efficient cancer control method in high-risk populations with the introduction of novel and promising HCC screening strategies based on various combinations of the most favourable biomarkers.

Additional biomarkers, such as tumour and non-tumour components from liquid biopsy, and metabolomic- and proteomic-based tools, are expected to emerge in the context of an increasingly personalized medicine, but they will require independent validation and strong evidence before being implemented in clinical practice.

## Figures and Tables

**Figure 1 ijms-24-04286-f001:**
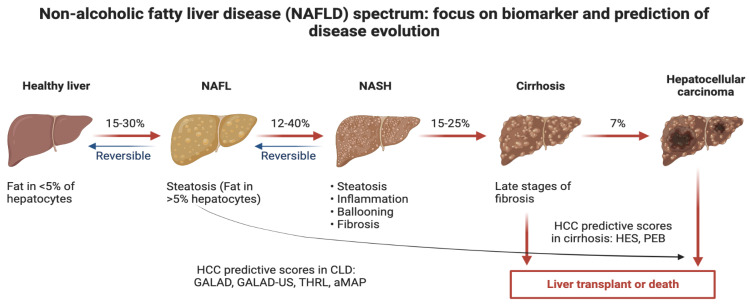
Depiction of a future possible scenario of prognostic scores in the context of hepatocellular carcinoma.

**Figure 2 ijms-24-04286-f002:**
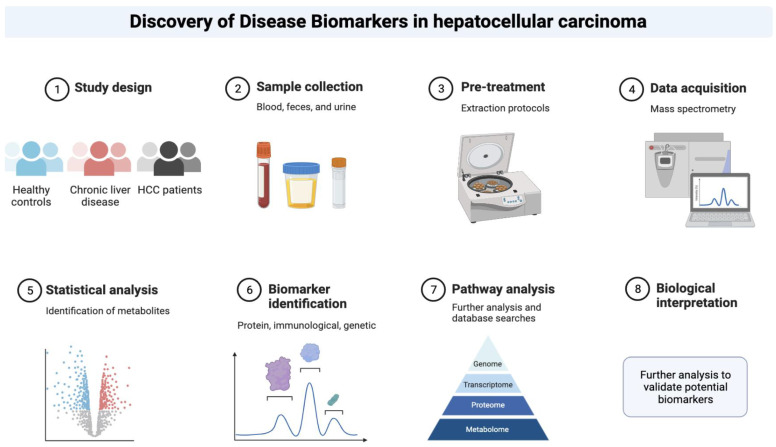
Depiction of the phases that could be taken to discover novel biomarkers in the context of hepatocellular carcinoma.

**Figure 3 ijms-24-04286-f003:**
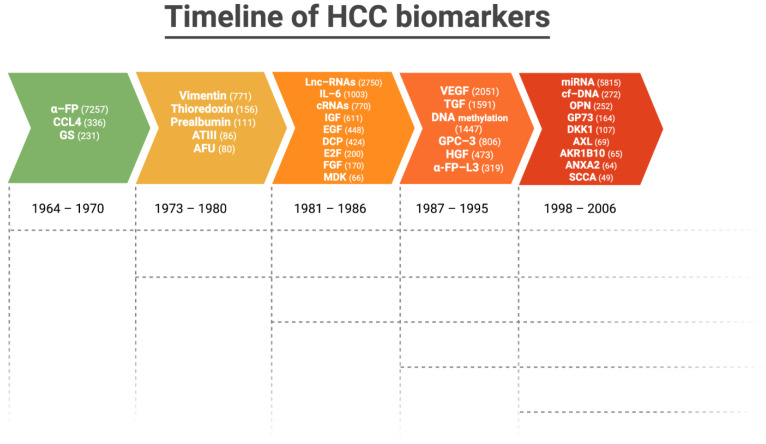
Timeline and number of articles (in parenthesis) published on the most validated biomarkers in hepatocellular carcinoma up to 2023.

**Table 1 ijms-24-04286-t001:** Example of candidate HCC diagnostic biomarkers. Sensitivity, specificity and AUROC shown as percentages (%).

Biomarker	Type	Study Type	Sensitivity (%)	Specificity (%)	AUROC (%)	Cut-Off	Ref.	Notes
α-FP	Protein	Systematic review	4–71	29–100	65	20–500 ng/mL	Tateishi, R. et al., 2008 [178]	Cirrhosis, Viral CLD
	Protein	Review	67.7	71	-	>100 ng/mL	Stefaniuk, P. et al., 2010 [19]	Cirrhosis any etiology (HCV and HBV prevalent), CLD
	Protein	Metanalysis	60–65	80–84	NA	20 ng/mL or 200 ng/mL	Colli, A. et al., 2021 [189]	CLD
α-FP-L3	Protein	Review	61.6	92	-	-	Stefaniuk, P. et al., 2010 [19]	Cirrhosis any etiology (HCV and HBV prevalent), CLD
	Protein	Metanalysis	34	92	75	3.84–10%	Zhou, J.M. et al., 2021 [190]	HCC
	Protein	Prospective	66.7–73.1	82,7	79–80	10%	Singal, A.G. et al., 2022 [10]	Cirrhosis
	Protein	Retrospective	64.3–67.4	94.5–95	64–69	10%	Tayob, N. et al., 2022 [36]	HCC
DCP	Protein	Metanalysis	67	92	89	4.5–7.5 ng/mL	Gao, P. et al., 2012 [43]	Cirrhosis any etiology, CLD
	Protein	Review	72.7	90	82	-	Stefaniuk, P. et al., 2010 [19]	Cirrhosis any etiology (HCV and HBV prevalent), CLD
GP73	Protein	Retrospective	62	88	-	12.69 relative units (HCC); 4.39 relative units (Cirrhosis)	Li, X. et al., 2009 [59]	Cirrhosis any etiology
	Protein	Prospective	87.1	83.9	92	7.4 relative units (HBV related HCC)	Hu, J.S. et al., 2010 [60]	CHB
	Protein	Metanalysis	76	86	88	2.36–400 relative units (HCC); 10–113.8 (early HCC)	Zhou, Y. et al., 2012 [61]	Cirrhosis, Viral CLD
GPC-3	Protein	Retrospective	84–85	92–95	85–89	-	El-Saadany, S. et al., 2018 [68]	HCC
OPN	Protein	Metanalysis	81	87	-	-	Sun, T. et al., 2018 [81,83]	CHB
DKK1	Protein	Prospective	87.3	82.9	82.6	8.92 ng/mL	Awad, A.E. et al., 2019 [94]	Cirrhotics any etiology (HCV prevalent)
	Protein	Metanalysis	71	87	85	-	Jiang X et al., 2020 [181]	-
AFU	Protein	Metanalysis	72	88	82	-	Gan Y et al., 2014 [97]	-
AXL	Protein	Retrospective	93.8	61.9	84	1243 pg/mL	Song X., et al., 2020 [100]	CLD any etiology and cirrhotics
MDK	Protein	Metanalysis	85	83	91	0.5 ng/mL	Zhang, Y., et al., 2020 [104]	HCC
	Protein	Systematic review and metanalysis	83.5	81.7	97	0.387–1.683 ng/mL	Lu, Q. et al., 2020 [105]	Viral CLD
	Protein	Cross-sectional	100	90	95	5.1 ng/mL	El-Shayeb, A.F. et al., 2021 [106]	Cirrhosis
AKR1B10	Protein	Prospective	81	61	76	1.51 ng/mL	Han, C., et al., 2018 [112]	Cirrhosis (HCV and HBV prevalent), CLD
ANXA2	Protein	Prospective	74	88	86	18 ng/mL	Shaker M.K., et al., 2017 [122]	CLD any etiology and cirrhotics, HBV, HCV and cryptogenetic; Early HCC
SCCA	Protein	Retrospective	41.9	82.6	70	3.8 ng/ml	Giannelli G et al., 2007 [126]	Cirrhosis any etiology, early HCC
SCCA-IC	Protein	Retrospective	52.3	75.7	67.5	104 AU/mL	Giannelli G et al., 2007 [126]	Cirrhosis any etiology, early HCC
GS	Protein	Retrospective	73.6	98.2	85	599.3 ng/mL	Liu P., et al., 2020 [134]	Cirrhosis HBV, ALD
Extracellular vesicles chip	Genetic	Retrospective	94.4	88.5	93	-	Sun N et al., 2020 [138]	Early HCC vs Cirrhosis
exo-miR-4746-5p	Genetic	Retrospective	81.8	91.7	95	-	Cho H.J. et al., 2020 [191]	Early HCC
exo-miR-10b-5p	Genetic	Retrospective	90.7	75	93	-	Cho H.J. et al., 2020 [191]	Early HCC
miR-320b, miR-663a, miR-4448, miR-4651, miR-4749-5p, miR-6724-5p, miR-6877-5p, and miR-6885-5p	Genetic	Retrospective	97.7	94.7	99	-	Yamamoto Y. et al., 2020 [192]	Any etiology
Circular RNAs	Genetic	Metanalysis	82	82	89	-	Nie, G. et al., 2022 [186]	Cirrhosis any etiology, CLD
cg04645914, cg06215569, cg23663760, cg13781744, and cg07610777	Genetic	Retrospective	-	-	95	-	Hlady RA., et al., 2019 [193]	HCC vs non tumour
Chemokines	Immunological	Retrospective	66 (CCL4); 71 (CCL5)	74 (CCL4); 78 (CCL5)	-	0.86 ng (CCL4) and 84 pg/mL (CCL5)	Sadeghi M et al., 2015 [165]	Cirrhosis any etiology, CLD
IL-6	Immunological	Metanalysis	46–73	73–95	72	3–12 pg/mL	Witjes CD et al., 2013 [170]	Any etiology

**Table 2 ijms-24-04286-t002:** Example of combinations of HCC diagnostic biomarkers. Sensitivity, specificity and AUROC shown as percentages (%).

Biomarker	Type	Study Type	Sensitivity (%)	Specificity (%)	AUROC (%)	Cut-Off	Ref.	Notes
α-FP+DKK1+OPN	Protein	Retrospective	87.50–88.76	88.37–88.70	94.8–94.9	6.79 ng/mL (α-FP), 1.31 ng/mL (DKK1), 15.11 ng/mL (OPN)	Ge, T. et al., 2015 [75]	CHB
α-FP+α-FP-L3+DCP	Protein	Prospective	85	85	75.86	-	Best, J. et al., 2016 [15]	CLD
α-FP+ DKK1	Protein	Prospective	87.5	92.3	94.6	-	Erdal, H. et al., 2016 [90]	Cirrhosis any etiology
α-FP+α-FP-L3+DCP	Protein	Prospective	85	85	75.86	-	Best, J. et al., 2016	CLD
α-FP+DCP	Protein	Metanalysis	84	86	89	-	Chen, H. et al., 2017 [178]	Cirrhosis, Viral CLD
α-FP+GPC-3	Protein	Retrospective	98.5	97.8		93	El-Saadany, S. et al., 2018 [68]	HCC
α-FP+α-FP-L3+DCP	Protein	Metanalysis	88	79	91	-	Wang, X. et al., 2020 [173]	CLD, Cirrhosis

**Table 3 ijms-24-04286-t003:** Example of candidate HCC diagnostic/prognostic algorithms. Sensitivity, specificity and AUROC shown as percentages (%).

Method	Study Type	Country	Sensitivity (%)	Specificity (%)	AUROC (%)	Ref.	Notes
APAC	Prospective	Germany	85.2	89.2	93	Lambrecht, J. et al., 2021 [213]	Cirrhosis any etiology
APAC	Retrospective	U.S.A.	85	89	92	Chalasani, N.P. et al., 2022 [144]	Cirrhosis any etiology
GAAP	Retrospective	China	87.2	79.2	91	Liu, M. et al., 2020 [207]	CLD
GALAD	Retrospective	United Kingdom	92	85	95	Johnson PJ et al., 2014 [204]	CLD
GALAD	Retrospective	Germany, Japan, and Hong Kong	80–91	81–90	85–95	Berhane, S. et al., 2016 [205]	CLD
GALAD	Retrospective	Germany, Japan, and United Kingdom	89.3	95.7	99	Schotten, C. et al., 2021 [208]	CLD (not viral)
GALAD	Retrospective	Germany, Japan, and United Kingdom	89.3	95.7	98	Schotten, C. et al., 2021 [208]	CHC
GALAD	Retrospective	Germany, Japan, and United Kingdom	76.9	95.4	96	Schotten, C. et al., 2021 [208]	CHB
GALAD	Prospective	U.S.A.	54.8–66.7	90	85	Singal, A.G. et al., 2022 [35]	CLD
GALAD	Prospective	U.S.A.	64–74	77.1–78.5	75	Tayob, N. et al., 2022 [36]	CLD
GALAD	Prospective	China	76	88	90	Huang, C. et al., 2022 [210]	CLD
GALAD-C	Retrospective	China	82.6	85.9	92	Liu, M. et al., 2020 [207]	CLD
GALADUS	Prospective	U.S.A.	95	91	98	Yang, J.D. et al., 2019 [212]	CLD
PEB	Retrospective	U.S.A.	63.6	-	-	Tayob, N. et al., 2018 [202]	HCV cirrhosis
HES	Retrospective	U.S.A.	52.56	48.13	-	Tayob, N. et al., 2019 [204]	Cirrhosis of any etiology
mt-HBT	Rerospective	U.S.A.	72	88	91	Chalasani, N.P. et al., 2022 [144]	CLD
Age+Gender+α-FP+PIVKA-II	Retrospective	China	83	90	93	Yang T et al., 2019 [95]	CLD

## Data Availability

Not applicable.

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
