# Peer review of "Updating the Clinical Application of Blood Biomarkers and Their Algorithms in the Diagnosis and Surveillance of Hepatocellular Carcinoma: A Critical Review"

_ijms, 2023, doi:10.3390/ijms24054286_

Round 1
Reviewer 1 Report
The study covers significantly biomarkers that are suggested to be increased in HCC patient, compared to non-cancerous counterparts in main and supplementary texts. This reviewer can ask whether the authors can have additional focus or summary on the biomarkers differential or common in HCC patients either from Western or Oriental countries. This can be interesting since there are a difference in 5 year survival rate and in GALAD score performance between the both parts (i.e., Germany/Italy vs China/Japan).
In addition, the authors can add certain references on tetraspan(in)s, such as TM4SF1 or TM4SF5 that are reported to be involved in HCC and NASH and . They can include PMID: 31876386, PMID: 29505836 for TM4SF1 and PMID 30956113, PMID 18357344, and PMID 34921636 for TM4SF5.
Author Response
We are especially grateful to the reviewer for providing us with the opportunity to improve our paper and for his very meaningful suggestion. As a result, we have included the necessary information on Western and Eastern studies in the discussion section. We have also included the cited references in the Supplementary material.

Reviewer 2 Report
In the review "Updating the application of blood biomarkers and their algorithms in the diagnosis and surveillance of Hepatocellular Carcinoma", the authors discuss the most used biomarkers and their role as prognostic indicators in the diagnosis of HCC. The authors have collected a large amount of information, but a critical analysis of the data presented is not a strong part of the work, although the authors claim that their review is critical.
It seems to me that the order of presentation of information in the review is neither convenient nor logical. For example, 2 figures, which serve as a basic introductory to the topic of the review, for some reason are inserted at the end, and not at the beginning of the text.
1) Figure 1 shows the main stages of HCC development. Obviously, it is necessary to begin the presentation of the material from this Figure. This will allow the authors to link the described studies with a specific stage of HCC.
2) Figure 2 illustrates the main stages of a classical study on the search and validation of prognostic biomarkers, regardless of the nature of the disease. Obviously, it is necessary to give this basic information at the beginning so that it can be appealed to when describing specific studies.
In addition, to strengthen the analytical component of the review, the authors should add a comparative analysis to the study of prognostic markers of protein and nucleic nature (Chapters 3.1 and 3.2) by adding to the text something like Table 1 in Chapter 4, in which predictive algorithms are compared. I would suggest, not even a table, but a figure, for example, a time line, where it is noted how many studies on a particular protein (nucleic) biomarker were published and when.
A similar Figure can be prepared for additional biomarkers placed in the supplement. Then it will become clear why such a division into more and less important markers was made.
Any additional tables or figures comparing the studies under discussion on any parameters are welcome!
Author Response
We appreciate the reviewer's valuable guidance. As indicated by the reviewer, we moved Figures 1 and 2 to the introduction section. We created a timeline for all of the biomarkers mentioned in the paper (and supplementary material), and provided the requested information in Figure 3 and Supplementary Figure 1, including a detailed description of the current studies on each biomarker. As a result, we transformed the presentation order slightly to prioritize the most influential, investigated, and validated biomarkers. In addition, we delivered the asked tables with information on the prognostic role of the major biomarkers, as documented in the new Tables 1, 2, and 3, as well as Supplementary Table 1.

Round 2
Reviewer 2 Report
I am absolutely satisfied with the corrections made. The Figures are signed a little unconventionally (from the top, not from the bottom). But this is more of an editorial rather than a scientific question.